# Tightening of tropical ascent and high clouds key to precipitation change in a warmer climate

Hui Su[1], Jonathan H. Jiang[1], J. David Neelin[2], T. Janice Shen[1], Chengxing Zhai[1], Qing Yue[1], Zhien Wang[3], Lei Huang[1,4], Yong-Sang Choi[5], Graeme L. Stephens[1] & Yuk L. Yung[6]

The change of global-mean precipitation under global warming and interannual variability is predominantly controlled by the change of atmospheric longwave radiative cooling. Here we show that tightening of the ascending branch of the Hadley Circulation coupled with a decrease in tropical high cloud fraction is key in modulating precipitation response to surface warming. The magnitude of high cloud shrinkage is a primary contributor to the intermodel spread in the changes of tropical-mean outgoing longwave radiation (OLR) and global-mean precipitation per unit surface warming ($dP/dT_s$) for both interannual variability and global warming. Compared to observations, most Coupled Model Inter-comparison Project Phase 5 models underestimate the rates of interannual tropical-mean $dOLR/dT_s$ and global-mean $dP/dT_s$, consistent with the muted tropical high cloud shrinkage. We find that the five models that agree with the observation-based interannual $dP/dT_s$ all predict $dP/dT_s$ under global warming higher than the ensemble mean $dP/dT_s$ from the ~20 models analysed in this study.

[1] Jet Propulsion Laboratory, California Institute of Technology, 4800 Oak Grove Drive, Mail Stop 183-701, Pasadena, California 91109-8099, USA. [2] Department of Atmospheric and Oceanic Sciences, University of California, Los Angeles, Los Angeles, California 90095, USA. [3] Department of Atmospheric Science, University of Wyoming, Laramie, Wyoming 82071, USA. [4] Joint Institute for Regional Earth System Science and Engineering, University of California, Los Angeles, Los Angeles, California 90095, USA. [5] Department of Environmental Science and Engineering, Ewha Womans University, Seoul 120-750, South Korea. [6] Division of Geological and Planetary Sciences, California Institute of Technology, Pasadena, California 91125, USA. Correspondence and requests for materials should be addressed to H.S. (email: Hui.Su@jpl.nasa.gov).

Precipitation is vital to life on Earth and regional precipitation changes accompanying anticipated global warming could exert profound impacts on ecosystems and human society. As a first-order constraint on the response of the hydrological cycle to climate change, the rate of global-mean precipitation ($P$) change per unit surface warming ($dP/dT_s$) is an essential measure that represents the sensitivity of the climate system to global warming. However, model predictions of $dP/dT_s$ under various warming scenarios vary from 1 to 3% K$^{-1}$ (ref. 1), resulting from model differences in radiative forcings and differences in responses to forcings. The responses include the fast response to direct forcings and the slow response mediated by the increase of surface temperature ($T_s$)[2,3]. The temperature-mediated $dP/dT_s$, termed hydrological sensitivity, ranges from 2 to 3% K$^{-1}$ (refs 3–5). Identification of the dominant processes that govern the intermodel spread in $dP/dT_s$ is critical for reducing the uncertainties of climate change predictions.

From the atmospheric energy budget point of view, on timescales of a year and longer, global-mean atmospheric latent heating (roughly $P$ multiplied by the latent heat of vapourization $L_v$) is approximately balanced by atmospheric column-integrated longwave cooling (LWC), shortwave absorption (SWA) and surface sensible heat exchange (SH)[6–10], that is,

$$L_v P = \text{LWC} - \text{SWA} - \text{SH}. \qquad (1)$$

On climatological means and for temporal variations, the magnitudes of LWC (signed positive for cooling the atmosphere) and its changes are more than twice of those for SWA and SH (signed positive for heating the atmosphere)[11,12]. Hence, global-mean precipitation is primarily determined by the rate of LWC[13,14] and the model disagreement in the LWC sensitivity to surface warming ($dLWC/dT_s$) contributes predominantly to the intermodel spread in $dP/dT_s$[5], although the model differences in the sensitivies of SWA, (refs 5,15) and SH (refs 3,16,17) to $T_s$ can also contribute substantially to the diversity in $dP/dT_s$.

A recent article[18] suggested that a stronger decrease of tropical high-altitude cloud fraction (CF) with increasing $T_s$ in a climate model could lead to a lower equilibrium climate sensitivity (ECS) and a higher hydrological sensitivity. They conjectured that the tropical high cloud shrinkage in response to surface warming, termed the iris effect analogous to a human's eye in response to varying light intensity[19], could result from enhanced convective aggregation in a warmer climate. Another study postulated that increased tropospheric static stability with surface warming could reduce the radiatively-driven clear-sky upper tropospheric convergence, leading to decreased anvil cloud amount[20], consistent with the fixed anvil temperature hypothesis[21,22]. However, it is not clear whether the high CF change is a dominant factor that determines the intermodel spread in $dP/dT_s$ in the models that participated in the Coupled Model Inter-comparison Project Phase 5 (CMIP5)[23] and what model physical processes are important for constraining the intermodel spreads in cloud and precipitation changes.

It is well known that the changes of cloud and precipitation are closely related to the changes of large-scale circulation[24–29]; however, no prior studies have demonstrated quantitatively the relation between circulation change and cloud amount change in terms of intermodel spreads. Such relations may shed light on the pathways towards improvements of model predictions of future climate change.

The schematic Fig. 1 illustrates the changes of the Hadley Circulation, tropical clouds, outgoing longwave radiation (OLR) at the top-of-atmosphere (TOA), and precipitation based on ensemble means of climate model simulations under global warming[29,30]. A prominent feature of the tropical circulation

change in a warmer climate is the intensification of zonal-mean equatorial ascent flanked by the weakening of upward motion to its north and south[29,30]. This feature is simulated by most climate models with varying magnitudes[29,31]. We term it the tightening of Hadley ascent (THA), in contrast to the well-known widening of the Hadley cell[32–34] as the latter highlights the poleward expansion of the descending branch of the Hadley Circulation even though the tightening and widening occur simultaneously under global warming[29,31]. A narrowing of the intertropical convergence zone (ITCZ) has been found in reanalyses data and precipitation observations from 1979 to 2014 (ref. 35). The physical mechanisms for the narrowing of the ITCZ are attributed to the advection of moist static energy by the zonal-mean circulation and the moist static energy divergence by transient eddies[31], broadly consistent with the upped-ante mechanism that emphasizes dry air advection from non-convective to convective regions[26]. These mechanisms do not explicitly involve the radiative effect of clouds.

On the other hand, the narrowing of equatorial ascending and cloudy regions corresponds to the expansion of dry radiator fins in the tropics[36], causing a greater longwave radiative loss to space. On the global mean, the atmospheric energy constraint requires increased latent heating to balance the enhanced atmospheric LWC. The increase of precipitation occurs primarily over the tightened convective zones near the equator, creating an intensified hydrological cycle with a 'wet get wetter and dry get drier' spatial pattern[25]. In addition, the direct response to increasing $CO_2$ could strengthen the subsidence over subtropical oceans and cause a further decrease of precipitation there[37], exacerbating the moisture and precipitation contrast between the climatologically wet tropics and dry subtropical regions. How the thermodynamically driven narrowing of the ITCZ interacts with the decrease of high cloud cover to govern global-mean precipitation change and the intermodel spread of hydrological sensitivity remains elusive.

In this study, we provide compelling evidence that the magnitude of the THA is closely related to the magnitude of the tropical high cloud shrinkage in response to surface warming across CMIP5 models. The latter drives the model differences in atmospheric LWC rate and thus global-mean precipitation sensitivity. We also show that the intermodel spreads in the interannual changes of the Hadley Circulation, high CF, OLR and precipitation per unit surface warming are highly correlated with those on the centennial timescale. As >70% of the models analysed here underestimate the interannual OLR and precipitation sensitivities, we infer that most CMIP5 models underestimate the hydrological sensitivity under global warming.

## Results

**Coupling between tropical circulation and high cloud changes.** The longwave radiative control on the global-mean precipitation change leads to high correlations between global-mean $dP/dT_s$ and $dLWC/dT_s$ across the models on both interannual and centennial timescales (Supplementary Fig. 1), while the inter-model spread in global-mean $dLWC/dT_s$ is primarily contributed by the spread in $dOLR/dT_s$ (see Supplementary Information). As high clouds have greater impact on atmospheric column LWC rate than clouds in the lower and middle troposphere[38] and the across-model spread in global-mean $dOLR/dT_s$ is largely driven by that in the tropics (20°S–20°N) (Supplementary Fig. 2), we focus on the tropical-mean high CF change. The high clouds refer to the clouds with tops at or above 440 hPa and their fractions are computed using a weighted average of high CFs under the maximum and random overlap assumptions (see Methods section and Supplementary Fig. 3).

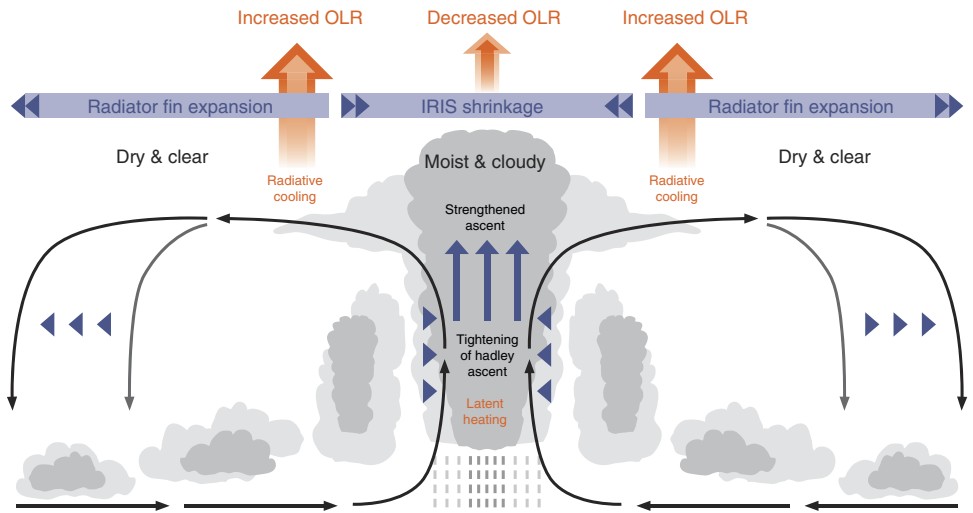

**Figure 1 | A schematic of model simulated changes in the Hadley Circulation and tropical clouds along with the OLR and precipitation changes in a warmer climate.** Black arrows mark the climatological Hadley Circulation with ascents near the equator and descents in the subtropics. Blue triangles indicate the poleward expansion of the descent zone and equatorward contraction of the ascending branch of the Hadley Circulation under global warming. The blue upward arrows indicate the strengthening of ascending motion over the equatorial tropics, where precipitation increases. The light and dark grey shadings of clouds correspond to the mean cloud distributions in the present and future climate, respectively, highlighting the rise of cloud top and the decrease of high cloud cover over the equatorial tropics and the reduction of low cloud amount in the subtropics. The higher cloud top causes decreased OLR over the equator, while the decrease of high cloud cover leads to increased OLR away from the equator. The schematic is based on multimodel-mean climate model simulations in response to increasing $CO_2$ (refs 29,30).

Shown in Fig. 2a and b, when the tropics is warmer on the interannual and centennial timescales, large-scale ascent strengthens and high CF increases over the deep tropics (10°S–10°N), but the opposite happens at the margins of convective zones. The increase of high clouds over the deep tropics is offset by the decrease of high clouds outside 10°S and 10°N, creating a rather small net change on the tropical means (Fig. 2c,d). On the tropical mean, most models produce negative regressions of high CF onto tropical-mean $T_s$ (Fig. 2c,d). This is in stark contrast to the high cloud and $T_s$ correlations on the local scale, which are mostly positive over the tropical oceans (Supplementary Fig. 4). Obviously, the interactions between high clouds and surface temperature do not occur just locally; instead, remote influence through tropical circulation change must be taken into account. It is known that high clouds are usually associated with upward motion, shown by the similar spatial patterns of vertical velocity and high CF regressions onto tropical-mean $T_s$ (Supplementary Fig. 5). We propose that the THA is the most pertinent property of the circulation change that is linked to the decrease of tropical-mean high CF in response to surface warming.

To reveal the magnitude of the tightening, the change of the areal fraction of upward velocity at 250 hPa over 20°S–20°N ($F_\omega$) per degree of surface warming ($dF_\omega/dT_s$) is calculated using the monthly model outputs for both interannual variations and the centennial changes (see Methods section). Then, we examine the relationship between the intermodel spread in $dF_\omega/dT_s$ and that in the tropical-mean $dCF/dT_s$ (Fig. 2c,d). The temperature-mediated $dF_\omega/dT_s$ is obtained as the regression slope of annual mean of monthly percentage of ascending areas within the tropics against the annual-mean tropical-mean $T_s$ in each model's abrupt4 × $CO_2$ experiment, and its correlation with the temperature-mediated $dCF/dT_s$ is shown in Supplementary Fig. 6a.

We found that ∼90% of the models (19 out of 21) produce a reduction of the tropical ascending area in response to increasing tropical-mean $T_s$ ($dF_\omega/dT_s$ being negative), consistent with the prediction from basic thermodynamics[24]. Moreover, the magnitudes of interannual and centennial $dF_\omega/dT_s$ across the

models are positively correlated with the changes of the width of the ascending branch of the Hadley Cell, define by the upward zonal-mean vertical velocity at 250 hPa (Supplementary Fig. 7). Although the weakening of the Walker Circulation may be an important component in the tropical circulation change on the interannual and centennial timescales[39], the THA appears to play a significant role in determining the tropical-mean upward mass flux change.

While most climate models simulate the THA, the magnitudes of the tightening differ substantially among the models. The intermodel spreads in the magnitudes of the circulation tightening and the high cloud shrinkage are well correlated for interannual variability and under global warming (Fig. 2c,d and Supplementary Fig. 6a). The models with a greater contraction of ascending areas tend to have a larger decrease of high cloud cover, and vice versa. Figure 2 and Supplementary Fig. 6a are clear manifestations of the intimate coupling between the THA and tropical high CF change in CMIP5 models. The across-model differences in the interannual sensitivities represent the diversity in the internal model physics that govern atmospheric circulation and cloud responses under the same prescribed observed sea surface temperature (SST) variations, while the across-model differences in the centennial sensitivities may include the effects of different SST warming patterns in the coupled simulations in addition to the different model physical parameterizations. It is expected that the large-scale circulation changes in the models are sensitive to convective parameterizations[40,41]. Because of the close relation between the circulation and cloud changes, model parameters that directly modify mass fluxes, for example, the entrainment rate in convective parameterizations, may have an important role in altering upper level cloud cover, as alluded to in a few earlier modelling studies[41,42]. Thus, constraining circulation-sensitive model physical parameters would likely help to reduce the across-model spread in high cloud amount. Note that the explained across-model variance in $dCF/dT_s$ by the tightening index is up to 42% ($R = 0.65$ for the 21 models on the long-term rates as CNRM_cm5 and INM_cm4 are missing RCP4.5 CF outputs). This is understandable as many other model

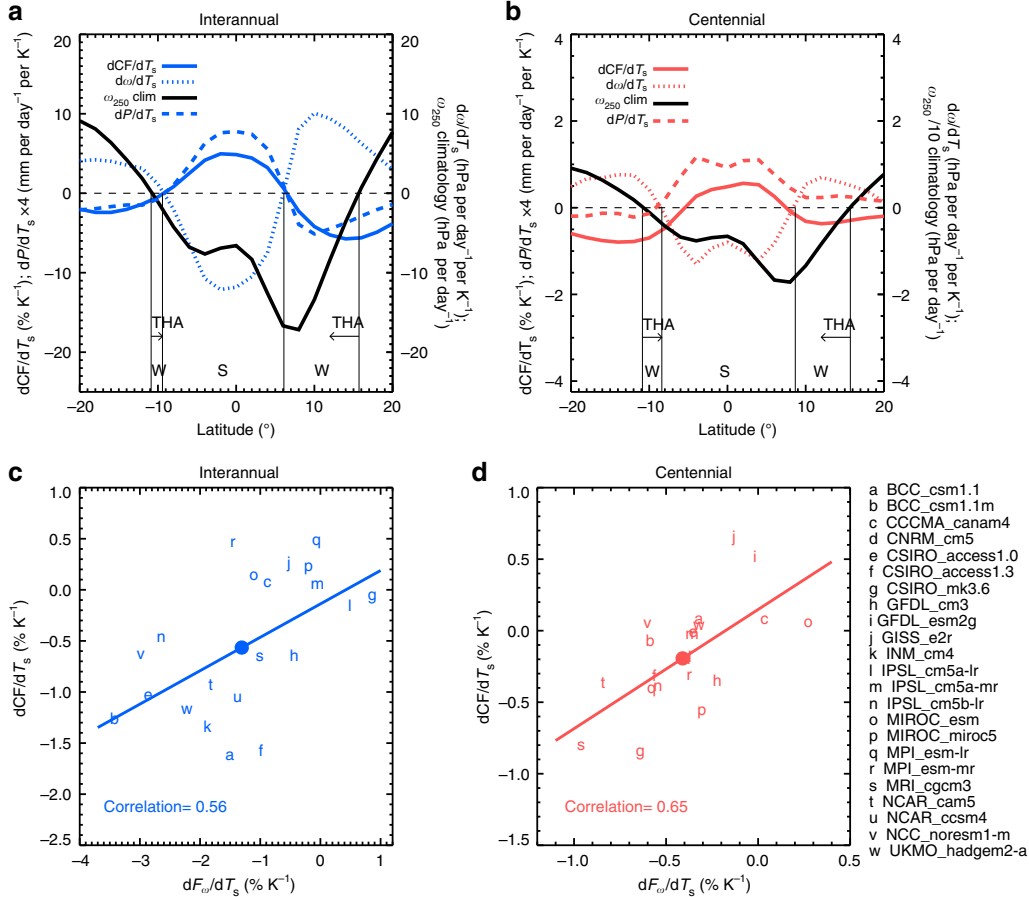

**Figure 2 | Relationship between tropical circulation and high CF sensitivities to surface warming.** (**a**) Interannual and (**b**) centennial multimodel-mean zonal-mean high CF (solid curves), precipitation (dashed curves) and vertical pressure velocity at 250 hPa ($\omega_{250}$, signed negative for ascending motion, dotted curves) changes per degree of tropical-mean (20°S–20°N) surface temperature increase. The multiyear-mean $\omega_{250}$ is shown in black solid curves. S and W indicate the strengthening and weakening segments of the Hadley ascent, and THA marks the tightening of Hadley ascent, defined by the weakening of upward motion at the flanks of the intensifying equatorial ascent. (**c**) Interannual and (**d**) centennial tropical-mean high CF change per unit surface warming $dCF/dT_s$ scattered against the change of the tropical ascending area fraction per unit surface warming $dF_{\omega}/dT_s$ for 21 CMIP5 models. The tropical ascending area is defined by $\omega_{250} < 0\,Pa\,s^{-1}$. Each model is represented by a lowercase letter. Multimodel means are marked in solid coloured circles. The least-squares linear regression lines and correlation coefficients between the x-axis and y-axis variables are shown.

deficiencies could affect the simulations of upper tropospheric CF. For example, the crude parameterizations of ice cloud microphysics[43] as well as poor representation of El Niño Southern Oscillation variability in the coupled simulations may also contribute to the different tropical circulation and cloud responses.

The remarkable resemblance between the interannual and centennial circulation and CF relations suggest that similar physical processes may be at work on both timescales. We note that the interannual and centennial sensitivities to surface warming are considered in a very simplistic way in this study: only the sensitivities to tropical-mean $T_s$ anomalies are analysed, regardless of the spatial patterns of SST warming. As the reduction of convective mass flux and the decrease of upper level clouds could happen under uniform SST warming based on fundamental thermodynamics[20,24], we speculate that the similarity between the interannual and centennial circulation and cloud relations may primarily result from the tropical-wide warming, common to both variabilities. However, as the climate models tend to produce an El Niño-like anomalous warming under the projected increase of greenhouse gases[44], it is not clear how much the SST warming patterns play a role in the similar interannual and centennial relations. On the other hand, it is

evident that El Niño is not a surrogate for global warming as the interannual and centennial sensitivities are different for each model, suggesting that factors beyond the tropical-mean $T_s$ anomalies affect the circulation and cloud changes.

As schematized in Fig. 1, the THA plays a central role in the dynamic and radiative controls on the precipitation change in response to surface warming. The large model spread in the extent of tightening is therefore a unique indicator for the model diversity in the global-mean precipitation change, confirmed by a strong negative correlation between the model spreads in centennial $dF_{\omega}/dT_s$ and $dP/dT_s$ ($R = -0.65$ for 23 models; Supplementary Fig. 8), and somewhat weaker interannual correlation (figure not shown). The coupled circulation and high cloud amount changes would also feedback onto $T_s$, but this is not the interest of this study.

The narrowing of tropical convective areas takes place simultaneously with the strengthening of equatorial ascent over the climatologically heavily precipitating regions, that is, the wet area. Thus, we expect the extent of the narrowing, $dF_{\omega}/dT_s$, would also be correlated with the magnitude of the precipitation increase over the wet area. We define the wet area as the grid boxes with monthly mean rain rates in the highest 15% of all tropical rain rates within 20°S–20°N. The corresponding threshold monthly

rain rate is $\sim 4$ mm per day in all CMIP5 models. The changes of the averaged rain rate for the wet area ($P_{wet}$) from the twentieth century to the twenty-first century normalized by the tropical-mean $T_s$ change are computed for each model. We find that the model spread in the centennial $dP_{wet}/dT_s$ is negatively correlated to the model spread in $dF_\omega/dT_s$ ($R = -0.52$ for 23 models; Supplementary Fig. 8) with a steeper slope than that for the linear fitting between $dF_\omega/dT_s$ and the global-mean $dP/dT_s$. This implies that the models with a stronger tightening would have a more severe 'wet get wetter' response under global warming than the models with a weaker tightening. This would have profound implications for regional extreme weather, floods and water resources.

**Dominance of high CF change in LWC.** The relationships between simulated tropical-mean (20°S–20°N) high CF and OLR sensitivities to $T_s$ across the models are displayed in Fig. 3. The intermodel spread in $dCF/dT_s$ is correlated with $dOLR/dT_s$ at $R = -0.77$ for interannual (excluding the outlier MRI_cgcm3 model) and $R = -0.71$ for centennial. The temperature-mediated $dOLR/dT_s$ and $dCF/dT_s$ are also correlated with $R = -0.71$ (Supplementary Fig. 6b). The model MRI_cgcm3 has an exceptionally large interannual $dCRE_{lw}/dT_s$ not related to its $dCF/dT_s$, probably because its prognostic ice crystal number concentration decreases significantly with surface warming in the present-day simulation[45]. Thus, this model is excluded when calculating the correlation coefficient in Fig. 3a.

A greater decrease in tropical high cloud cover leads to a greater loss of longwave radiation at the TOA. The intermodel spreads in $dCF/dT_s$ on both timescales are significantly correlated with the spreads in $dOLR_{clr}/dT_s$ and $dCRE_{lw}/dT_s$ (Supplementary Fig. 9). The decrease of high cloud amount reduces the cloud longwave warming effect on the Earth-atmosphere system, enlarging the dry and clear areas through which lower tropospheric thermal emissions escape to space and enhancing the negative longwave radiative feedback. However, we found that the intermodel spread in $dCF/dT_s$ is not a primary contributor to the spread in ECS because other feedback processes such as the shortwave cloud feedbacks from low-level clouds might overcome the radiative effects of high clouds and the shortwave effects of high cloud changes tend to cancel their longwave effects

(Supplementary Fig. 10). Although the strongly negative correlations between the model spreads in $dCF/dT_s$ and $dOLR/dT_s$ are consistent with our existing knowledge, they stress the predominance of high cloud coverage in driving the model diversity in the longwave radiative feedback and thus global-mean precipitation sensitivity. Other factors, such as cloud top height, cloud emissivity, cloud optical thickness and upper tropospheric water vapour, all could influence the rate of OLR change with surface temperature and contribute to the model differences in the short- and long-term longwave radiative feedback strength. Our analysis shows that the model differences in the tropical-mean high CF sensitivity to surface warming account for $\sim 50$–60% of the across-model variances of $dOLR/dT_s$ for interannual and centennial variations. In comparison, the model differences in the rate of upper troposphere water vapour change with surface temperature have a relatively small contribution to the intermodel spread in $dOLR/dT_s$ (see Supplementary Information).

In Fig. 3, the observed $dCF/dT_s$ and $dOLR/dT_s$ on the interannual timescale averaged over 20°S–20°N using the best available satellite data sets are marked in comparison with the modelled rates. Owing to different instrument cloud detection limitations, the observed tropical-mean high CF sensitivities vary from $-2.4$ to $-1.4\% \, K^{-1}$ (Supplementary Fig. 11 and Supplementary Table 2), which are similar to the unnormalized values reported in Lindzen et al.[19] based on daily cirrus CF over the Western Pacific but $\sim 10$ times smaller than the normalized $-22\% \, K^{-1}$ for the original iris effect[19]. Although various satellite data sets produce different magnitudes of $dCF/dT_s$, the values are all negative, that is, tropical-mean high cloud cover tends to decrease when surface temperature increases. We note that the interannual CF sensitivity includes the cloud response to the increase of tropical-mean $T_s$ as well as the response to the change of SST shape. Using different temporal periods or removing strong El Niño events in the time series yields somewhat different $dCF/dT_s$ but does not affect the results qualitatively. These satellite observations suggest there exists a mechanism for the tropical high cloud shrinkage on the interannual timescale.

We recognize that accurate evaluations of the simulated high CF variations with observations would require utilizing satellite

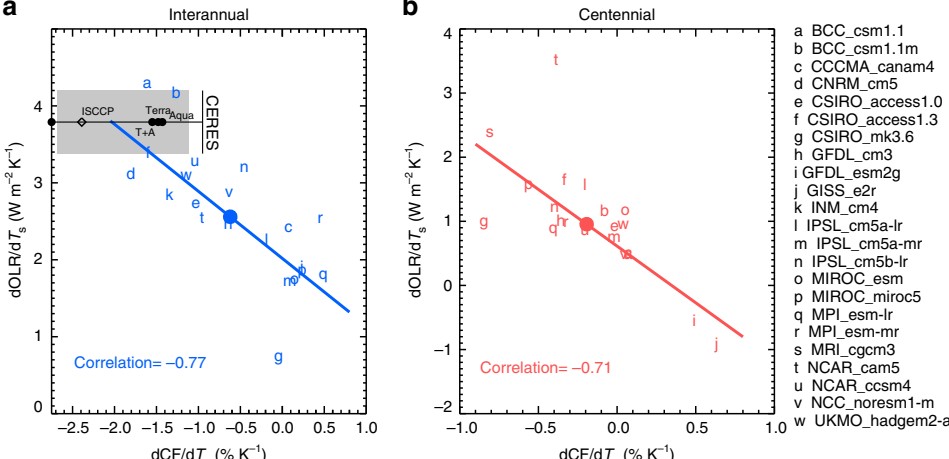

**Figure 3 | Relationship between tropical high CF and OLR sensitivities to surface warming.** (**a**) Interannual and (**b**) centennial tropical-mean OLR change per unit surface warming $dOLR/dT_s$ scattered against tropical-mean high CF change per unit surface warming $dCF/dT_s$ for 21 CMIP5 models. Each model is represented by a lowercase letter. Multimodel means are marked in solid coloured circles. The least-squares linear regression lines and correlation coefficients between the x-axis and y-axis variables are shown. The observed $dCF/dT_s$ from multiple satellite sensors along with the observed $dOLR/dT_s$ from CERES EBAF are shown in black symbols in **a**. The grey-shaded area marks the uncertainties of the observed data, based on 95% confidence interval of the regression slope between deseasonalized CERES EBAF OLR and HadCRUT4 $T_s$ and the range of the observed $dCF/dT_s$ from the satellite data used.

simulators in each model. Therefore, the observed $dCF/dT_s$ values are used here mainly to aid the interpretation of the biases in modelled longwave radiative feedback. Compared to the observed $dCF/dT_s$, the multimodel mean of $-0.7\% \, K^{-1}$ is significantly lower than the observations. The observed $dOLR/dT_s$ based on the Clouds and the Earth's Radiant Energy System (CERES) data from March 2000 to October 2015 is $3.8 \pm 0.4 \, W \, m^{-2} \, K^{-1}$, which is more robustly measured by satellite instruments than $dCF/dT_s$. Using the combined radiative fluxes from the Earth Radiation Budget Experiment (ERBE) for 1995–1999 and CERES for 2000–2005, we obtain a similar rate of $dOLR/dT_s$ at $4.0 \pm 0.5 \, W \, m^{-2} \, K^{-1}$. Both rates are consistent with previous studies[18,46,47]. All models except BCC_csm1.1 and BCC_cms1.1m underestimate the magnitude of $dOLR/dT_s$ as found previously[18,47]. If a model simulates a strong decrease of high CF with surface warming, its $dOLR/dT_s$ would be relatively large based on the negative correlations shown in Fig. 3. The observed CF and OLR sensitivities are within the scatter of the individual models relative to the linear regression line in Fig. 3a.

Consistent with muted high cloud shrinkage in the models, the simulated upper tropospheric moistening with surface warming is generally overestimated (Supplementary Fig. 12). However, using the water vapour radiative kernels[48,49], we find the multimodel-mean moist bias of $2\% \, K^{-1}$ in the upper troposphere would contribute a low bias on the order of $0.05 \, W \, m^{-2} \, K^{-1}$ to $dOLR/dT_s$, only a small fraction of the total model biases. And the intermodel spread in $dOLR/dT_s$ is not significantly correlated with that of the upper tropospheric moistening rate (see Supplementary Information).

**Constraining the hydrological sensitivity using observations.** Last, we explore the implications of the low biases in the tropical high cloud shrinkage and longwave radiative feedback on the model predictions of future global-mean precipitation change. Here we use the temperature-mediated global-mean precipitation change per degree of surface warming derived from the linear regression between annual-mean precipitation and surface temperature in the abrupt4 × $CO_2$ experiments as a 'clean' measure of the hydrological sensitivity[1,3–5] (Supplementary Table 1). This measure excludes the fast precipitation response to direct $CO_2$ forcing, which is independent of $T_s$ change[1,2]. The centennial precipitation change per unit surface warming is also analysed, representing the end of the twenty-first century total precipitation change relative to the present-day climate given transient $CO_2$ increase and other radiative forcings (Supplementary Fig. 13).

As shown in Fig. 4a, a strong correlation exists between the interannual tropical-mean $dOLR/dT_s$ and the interannual global-mean $dP/dT_s$ ($R = 0.68$ for 23 models), consistent with the longwave radiative control on precipitation. The approximate linear relation between the interannual $dOLR/dT_s$ and $dP/dT_s$ across the 23 models allows us to obtain an estimate of the interannual $dP/dT_s$ constrained by the observed tropical-mean $dOLR/dT_s$, independently of the interannual $dP/dT_s$ derived directly from the the linear regression between the Global Precipitation Climatology Project (GPCP) global-mean precipitation and the Hadley Centre and Climate Research Unit 4.4.0.0 surface temperature over the period of 1995–2005 (Supplementary Fig. 14).

Given the observed tropical-mean $dOLR/dT_s$ at $3.8 \pm 0.4 \, W \, m^{-2} \, K^{-1}$, we estimate the OLR-constrained interannual global-mean $dP/dT_s$ to be within 1.0 and 2.6 $W \, m^{-2} \, K^{-1}$ at the 95% confidence level (marked by grey horizontal lines in Fig. 4a) (see Supplementary Information). Combined with the interannual precipitation sensitivity from GPCP at $2.7 \pm 0.9 \, W \, m^{-2} \, K^{-1}$ (marked by the open black circle with grey

shading in Fig. 4a), the likely interannual global-mean $dP/dT_s$ at the 95% confidence level is between 1.8 and 2.6 $W \, m^{-2} \, K^{-1}$, determined by the overlapped range for the OLR-constrained and GPCP-derived precipitation sensitivities.

Furthermore, we find that the interannual global-mean $dP/dT_s$ is highly correlated with the temperature-mediated $dP/dT_s$ ($R = 0.64$ for 21 available models) and the centennial $dP/dT_s$ ($R = 0.55$ for 20 available models; Supplementary Fig. 13). The approximately similar correlations imply that the intermodel spread in the centennial $dP/dT_s$ is largely driven by the model differences in the temperature-mediated $dP/dT_s$, although the fast response to direct $CO_2$ forcing reduces the magnitudes of the tempeature-mediated $dP/dT_s$ by varying extent. Similar correlations are found between interannual and temperature-mediated $dOLR/dT_s$ or between interannual and centennial $dOLR/dT_s$ (Supplementary Fig. 15), but not for $dF_\omega/dT_s$ or $dCF/dT_s$, probably because the circulation and CF sensitivities are more dependent on the mean states and the temporal scales of variabilities. Hence, we focus on applying observed $dOLR/dT_s$ and $dP/dT_s$ as emergent constraints on the predictions of long-term precipitation sensitivity.

Using the observation-based interannual $dP/dT_s$ to identify the better performing models, we find only five models that simulate the observed interannual $dP/dT_s$ within the 95% confidence level (Fig. 4b). All the five models have temperature-mediated $dP/dT_s$ higher than the ensemble mean of 2.26 $W \, m^{-2} \, K^{-1}$ (2.6% $K^{-1}$) from the 21 models. The mean temperature-mediated $dP/dT_s$ from the five models is 2.35 $W \, m^{-2} \, K^{-1}$ (2.7% $K^{-1}$). Their across-model standard deviation is 0.08 $W \, m^{-2} \, K^{-1}$, 66% smaller than the standard deviation from the 21 models. Compared to the original values of the temperature-mediated hydrological sensitivity from 2.1 to 3.2% $K^{-1}$, the observation-constrained predictions of the hydrological sensitivity range from 2.6 to 2.9% $K^{-1}$, all in the higher end of the model ensemble.

For centennial $dP/dT_s$, the mean of the five better-performing models is slightly higher than the mean of all models; however, there is a large spread among the five models, indicating the model diversity in the fast response to direct $CO_2$ forcing. Nevertheless, the statistically significant positive correlation between the 20 models' interannual and centennial $dP/dT_s$ suggests that the models that simulate a stronger interannual $dP/dT_s$ tend to produce a greater $dP/dT_s$ at the end of the twenty-first century. Using the value of $dOLR/dT_s$ from the combined ERBE and CERES data for the period of 1995–2005 yields a slightly higher upper limit of interannual OLR-constrained $dP/dT_s$, but it does not materially change our conclusions.

The upward shift of ensemble mean $dP/dT_s$ is opposite to the effects of constraining models' solar absorption by water vapour[5]. It clearly demonstrates that climate models have compensating errors in simulating the interactions between circulation, cloud, radiation and precipitation. Emergent constraints on precipitation change based on a particular aspect of the hydrological processes may be biased. It is important to examine the precipitation changes from multiple perspectives and apply a variety of observational metrics to evaluate the models and guide model improvements.

## Discussion
By analysing the intermodel spreads in precipitation sensitivity and associated dynamic, thermodynamic and radiative quantities, we show that the model differences in simulating the extent of the tightening of the ascending branch of the Hadley Circulation in a warmer climate are highly correlated with the model spreads in high CF sensitivity, the rate of longwave radiative cooling and global-mean precipitation change for both interannual variability and long-term climate change. The dynamic and radiative processes are intimately coupled to produce the intensification

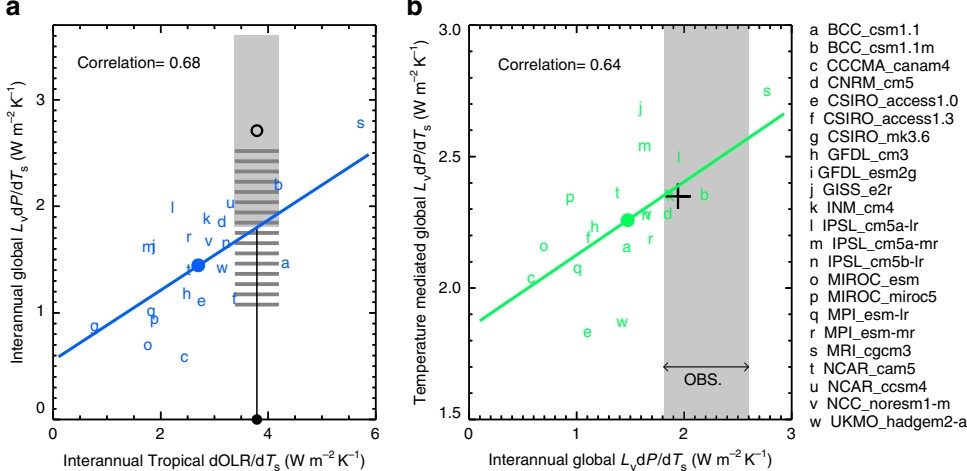

**Figure 4 | Emergent constraint on the hydrological sensitivity.** (**a**) Interannual global-mean precipitation change per unit surface warming $L_v dP/dT_s$ scattered against interannual tropical-mean OLR change per unit surface warming $dOLR/dT_s$ for 23 CMIP5 models. The CERES $dOLR/dT_s$ (the black dot on the x-axis) with the 95% confidence level marked horizontally in grey shading. The GPCP global-mean $L_v dP/dT_s$ is shown in open black circle with the 95% confidence level marked vertically in grey shading. The CERES OLR-constrained $L_v dP/dT_s$ estimate based on the linear regression relation between the interannual $L_v dP/dT_s$ and $dOLR/dT_s$ is marked by grey horizontal lines with their length corresponding to the 95% confidence level. The overlapped range of the GPCP and CERES OLR-constrained $L_v dP/dT_s$ is used as the best estimate of observational interannual $L_v dP/dT_s$ with the 95% confidence level. (**b**) The temperature-mediated global-mean $L_v dP/dT_s$ scattered against the interannual $L_v dP/dT_s$ for 21 CMIP5 models. The best estimate of the observational interannual $L_v dP/dT_s$ is marked in grey shading. Each model is represented by a lowercase letter. The ensemble model means for the 21 models and the five better-performing models are shown in solid circles and black cross, respectively. The least-squares linear regression lines and correlation coefficients between the x-axis and y-axis variables are shown.

of the hydrological cycle. The narrowing and strengthening of the equatorial ascent is a key contributor in this feedback loop on both timescales. Constraining circulation-sensitive model parameters would be one effective pathway towards improving upper level cloud simulations, critical for reducing the uncertainties of hydrological sensitivity, although the high cloud amount changes have no simple relation with the models' climate sensitivity. Moreover, as circulation and cloud changes are highly coupled, model parameterizations that directly affect high cloud formation and evolution could also impact large-scale circulation. Thus, understanding the complex interactions between circulation and cloud changes would be of utmost importance for accurate climate change predictions.

Our analysis reveals that the relative magnitudes of the OLR and precipitation sensitivities to surface warming vary consistently across the models on the interannual and centennial timescales, allowing us to constrain the likely range of future precipitation change based on short-term observations. However, there are noticeable differences between the interannual and centennial sensitivities for each model as El Niño is not a surrogate for global warming. In particular, the effects of the SST warming patterns on the tightening of Hadley ascent and the tropical high cloud shrinkage merit further investigation.

Using the observed interannual tropical-mean $dOLR/dT_s$ and global-mean $dP/dT_s$ to constrain the models, we can effectively reduce the model spread in the hydrological sensitivity by a factor of 3. The models that agree with the observed interannual $dP/dT_s$ at the 95% confidence level predict that the global-mean precipitation would increase with $T_s$ at a rate between 2.6 and 2.9% K$^{-1}$, all on the upper half of the CMIP5 model ensemble. The underestimates of the high cloud shrinkage with surface warming in most CMIP5 models also have profound implications for the regional precipitation change, whereas the magnitude of the intensification of extreme precipitation in climatologically heavily precipitating regions might likely be on the higher end of current model predictions.

## Methods

**Models.** We employ 23 climate model simulations driven by the observed SST and the corresponding coupled model simulations from historical and RCP4.5 scenarios (Supplementary Table 1) available at the CMIP5 archive (http://cmip-pcmdi.llnl.gov/cmip5/). Two models (CNRM_cm5 and INM_cm4) do not have CF outputs in the historical runs so that only 21 models are used where applicable. The available abrupt$4 \times CO_2$ and piControl experiments for these models are also analysed. The ECS values for the models are taken from Su *et al.*[29] and Mauritsen and Stevens[18].

We are interested in both short-term variabilities and long-term climate changes. The short-term variabilities refer to the interannual variations in present-day climate. We use the simulations driven by observed SST so that only atmospheric processes are taken into account (similar to the Atmospheric Model Inter-comparison Project, AMIP). Previous studies offer some promise that the interannual variations may bear imprints of long-term climate feedbacks[50–53], although caveats exist[53]. The short-term sensitivity of each variable to surface warming is derived from the regression slope of the deseasonalized anomalies against the $T_s$ anomalies. A 5-month running mean is applied to all monthly anomalies to reduce noises. We analyse the interannual relations for the decade of 1995–2005, consistent with Mauritsen and Stevens[18]. This is the period without large volcanic activities and the correlation between precipitation and surface temperature on the interannual timescale is less scattered than the previous periods[54]. Similar analyses were performed using the coupled historical simulations, but the relationships between high CF, radiation and precipitation were much noisier in the coupled simulations than the uncoupled AMIP simulations.

The long-term climate changes refer to the centennial differences of global or tropical averaged quantities between the 25-year climatological means in the twenty-first and twentieth centuries (the 2074–2098 averages from the RCP4.5 runs minus the 1980–2004 averages from the historical runs) normalized by the differences in global-mean or tropical-mean $T_s$. Note that such long-term sensitivities include both fast responses to direct $CO_2$ forcing and slow temperature-mediated responses. The temperature-mediated precipitation changes per unit surface warming derived from the abrupt$4 \times CO_2$ experiments are treated as the 'clean' measure of the hydrological sensitivity and are based on the linear regression slopes between the annual-mean global-mean precipitation anomalies against $T_s$ anomalies in the first 150 years of the simulations. The annual-mean anomalies are with respect to the 21-year averaged climatological values from the piControl experiments centred on the correspond year in the abrupt$4 \times CO_2$ experiments, consistent with DeAngelis *et al.*[5]. The temperature-mediated tropical-mean $dF_\omega/dT_s$, $dCF/dT_s$ and $dOLR/dT_s$ are computed similarly. We note that our definitions of sensitivities consist of the responses to surface warming and their feedbacks onto $T_s$. The coupled relationships, rather than one-way cause and effect, are applicable to all the sensitivities presented in this study.

In this study, high clouds pertain to the clouds with tops at or above 440 hPa altitude, regardless of cloud optical thickness. The CMIP5 models output CF at models' vertical levels and only limited models produce satellite simulator CFs. Therefore, we compute total high CF for each model using the same approach to obtain sufficient number of models for both short- and long-term analyses. As the maximum overlap assumption tends to underestimate the total CF while the random overlap assumption tends to overestimate the total CF (Supplementary Fig. 3), we use weighted averages of the high CFs computed under the maximum and random overlap assumptions separately. We found that the weights of 2/3 for maximum and 1/3 for random overlap CFs yield a close match in both total amount and temporal variations to the International Satellite Cloud Climatology Project (ISCCP) simulator CFs on the tropical averages (20°S–20°N) in three AMIP models that output ISCCP simulator CF results (Supplementary Fig. 3). Hence, the same weighted averages of maximum and random overlap CFs are used to represent the total high CFs in all models. Our conclusions are not sensitive to the exact weights for either overlap assumption.

For high CF and OLR sensitivities, we focus on the tropics between 20°S and 20°N, consistent with Mauritsen and Stevens[18] because this latitudinal band encompasses deep convective clouds and their anvils, central to the debate involving the iris hypothesis[19,55,56]. For precipitation, global mean and tropical wet area mean are examined.

**Observations.** The Hadley Centre and Climate Research Unit surface temperature 4.4.0.0 data set (http://www.metoffice.gov.uk/hadobs/hadcrut4) is used[57]. We use radiative fluxes measured by the CERES and various satellite retrievals of high CF in the past 15 years (from March 2000 to October 2015) in relation to $T_s$ to obtain observational estimates of short-term longwave radiative feedback and high cloud sensitivity to $T_s$ (Supplementary Table 2). The CERES-EBAF version 2.8 data[58] can be obtained at http://ceres.larc.nasa.gov. The tropical OLR sensitivity to for the 1995–2005 period was examined using the combined ERBE[59] and CERES data. While the CERES radiative flux measurements at the TOA have high accuracy and stability[60], the high CF retrievals from different satellite missions have considerable uncertainties due to instrument sensitivity and difficulties in identifying cloud top height for nadir-viewing passive sensors. The length of satellite record also affects the derived CF sensitivities (Supplementary Table 2). The observed high CF measurements include those from the Collection 6 MODerate-resolution Imaging Spectroradiometer (MODIS) on Terra (from July 2002 to June 2010) and Aqua (from July 2002 to June 2015) satellites[61] (http://dx.doi.org/10.5067/MODIS/ MOD08_M3.006), Atmospheric Infrared Sounder (AIRS) on Aqua[62] (from September 2002 to December 2013; https://disc.gsfc.nasa.gov/AIRS) and joint CloudSat/CALIPSO (Cloud-Aerosol Lidar and Infrared Pathfinder Satellite Observations) retrieval[63,64] (from June 2006 to June 2015, http://www.cloudsat. cira.colostate.edu/data-products/level-2b/) as well as the bias-corrected high CF data from the ISCCP[65,66] (from January 1995 to December 2005; https://eosweb. larc.nasa.gov/project/isccp/isccp_d2_table). The Terra MODIS CF retrieval after 2010 is found to have a high bias over ocean and thus not included in the regressions[67]. The joint CloudSat/CALIPSO high CF retrieval is based on the 2B-CLDCLASS-LIDAR product, which averages CALIPSO lidar measurements to CloudSat footprints and combines with CloudSat measurements to compute the ratio of cloud pixels with cloud top at or above 440 hPa to the total number of measurements. The AIRS effective CF accounts for both cloud areal coverage and cloud emissivity; hence, its variability may deviate from that of actual CF.

The observed precipitation during the period of 1995 to 2005 from the GPCP[68] (https://precip.gsfc.nasa.gov/gpcp_v2.2_data.html) and the CPC (Climate Prediction Center) Merged Analysis of Precipitation[69] (https://www.esrl.noaa.gov/ psd/data/gridded/data.cmap.html) data sets are analysed. The CPC (Climate Prediction Center) Merged Analysis of Precipitation precipitation sensitivity was not used for the observed precipitation sensitivity due to the large uncertainty (Supplementary Fig. 14). To better understand the model biases in longwave radiative feedback, combined upper tropospheric water vapour measurements from Aqua AIRS (440–250 hPa; https://disc.gsfc.nasa.gov/AIRS) and Aura Microwave Limb Sounder (250–100 hPa)[70] (https://mls.jpl.nasa.gov/products/h2o_product.php) are examined and compared to the model results (Supplementary Fig. 12). The uncertainty of the Microwave Limb Sounder/AIRS water vapour data is about 25% in the upper troposphere.

For satellite measurements, we use their respective available periods closest to the period of 1995 to 2005. Deseasonalized monthly anomalies are used to calculate the sensitivities to $T_s$. A 5-month running mean is performed to achieve a meaningful correlation for the interannual variabilities. The derived interannual sensitivities to $T_s$ and the periods used for each data set are listed in Supplementary Table 2. The variabilities indicate the 95% confidence intervals on the regression slopes.

The observed $dCF/dT_s$ from various satellite sensors exhibit large differences (Supplementary Fig. 11). The ISCCP bias-corrected high CF shows the steepest decrease per unit surface warming, $-2.39 \pm 0.29\%\,K^{-1}$. The Terra and Aqua MODIS high CFs yield negative rates of $-1.48$ and $-1.43\%\,K^{-1}$, respectively, while the combined Terra and Aqua MODIS measurements produce a decrease of high cloud cover at the rate of $-1.55\%\,K^{-1}$. The AIRS effective high CF varies with surface temperature at the rate of $-0.68 \pm 0.24\%\,K^{-1}$; however, as this represents combined effects of cloud emissivity and cloud coverage, we do not

include the AIRS results in Fig. 3a. The joint CloudSat/CALIPSO daytime retrieval of high CF from 2006 to 2015 yields a rate of $-0.88 \pm 2.41\%\,K^{-1}$ (not shown in Fig. 3a). The large uncertainty is partly due to the different instrument sensitivity and limited spatial sampling of the active sensors. The CloudSat radar operates only in the daytime after April 2011 and the quality of the data may be adversely affected. Excluding the data after April 2011 results in a very short record and poor correlation between the CF and $T_s$ anomalies.

**Data and code availability.** The data generated from the public accessible CMIP5 model outputs and satellite observations and the code used during the study are available on request from the authors. Please contact the corresponding author at Hui.Su@jpl.nasa.gov.

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

## Acknowledgements

We acknowledge the funding support from NASA NEWS, AST, MAP, NDOA and NSF. We greatly appreciate Michael Wong in making the schematic figure. We thank Shang-min Long, Ryan Stanfield and Jung-Min Park for assistance in some parts of the auxillary analyses. We thank Drs Brian Soden and Karen Shell for providing the radiative kernel functions. We appreciate helpful discussions with Drs Chris Bretherton, Anthony DeAngelis, Feifei Jin, Xin Qu and Shang-Ping Xie. We thank three anonymous reviewers for insightful suggestions. This work was performed at Jet Propulsion Laboratory, California Institute of Technology, under contract with NASA.

## Author contributions

H.S. designed the analysis and wrote the paper. H.S., J.H.J., T.J.S. and C.Z. analysed the CMIP5 model simulations and observations. Q.Y. analysed the MODIS cloud fraction data. Z.W. and L.H. analysed the joint CloudSat/CALIPSO cloud fraction retrieval. Y.-S.C. analysed the combined ERBE-CERES data. J.D.N., G.L.S. and Y.L.Y. provided suggestions for the analysis and comments on the manuscript. Everyone edited the manuscript.

## Additional information

**Competing interests:** The authors declare no competing financial interests.

