## [Peer Review File · Nature Communications]

Reviewers' comments:

Reviewer #1 (Remarks to the Author):

Review of: Tightening of Hadley ascent key to radiative control on precipitation change under warmer climate.

This paper combines two aspects. One is an observational analysis that shows a negative correlation between high cloud cover and sea surface temperature in the tropics from 20N-20S, and a strikingly large positive correlation between OLR and surface temperature. Another is an analysis of models that compares the sensitivity of high cloud fraction and outgoing longwave radiation to surface temperature. By combining these two things the authors imply that the reduction of high cloud fraction with warming increases the cooling of the atmosphere and thereby controls the rate at which global mean precipitation increases with surface temperature. By comparing the observed covariance between cloud fraction and surface temperature in observed month-to-month variability with model results, the authors conclude that the sensitivity of precipitation to temperature should be greater in models than it is currently represented to be.

I have two primary objections to the research. First: the applying the observational analysis to climate sensitivity is inappropriate, so that the presentation of this analysis as an emergent constraint is not yet warranted. It is very naive. The values of some covariances are outside the range expected from decades of research, so their interpretation as a climate sensitivity constraint is highly questionable. Second: the model analysis is not new, but has been described before.

1) In the model analysis the authors correlate the deviations of monthly mean cloud fraction anomalies and OLR anomalies with monthly surface temperature anomalies. It is not demonstrated what produces these anomalies and whether they represent a response of clouds to SST or a response of SST to clouds. Therefore to characterize them as useful in a feedback constraint context is not warranted without much further research. One could hypothesize that these changes are associated with natural variability of the amount of tropical deep convection averaged over the tropics. An example of this is the Madden Julian Oscillation which produces coordinated changes in high cloud fraction and SST over a large enough area to affect the whole tropics. This would occur in both summer and winter. When the SST is high clouds are more likely to form. When they form they cool the surface through reflection of solar radiation and through enhanced vertical transport of energy. Thus SST and cloud fraction can be negatively correlated. If this is the cause of the month-to-month variability, then it is not measuring the response of clouds to SST in a climate feedback sense, but rather a combination of clouds forcing SST and SST forcing clouds in disequilibrium oscillations. This might be expected to produce a larger negative correlation between cloud fraction and SST than one might get in response to global climate change. Therefore, I don't think it should be used to gauge the reliability of model simulations of climate change, and should not be used as a constraint, until it can be more deeply studied and shown to be a good analog for the equilibrium response to surface warming.

The sensitivity of OLR to surface temperature in line 164 is about twice the value obtained in numerous observational and theoretical studies. This suggests that the change of SST driven by the cloud cover is contributing to this value. The simple, nearly blind comparison of monthly, tropical mean anomalies of OLR, high cloud, and SST done in this paper is not adequate to overturn decades of research. Most models do not simulate the MJO properly, and that might be a reason why the short-term variability in those models cannot produce the very large value inferred from the CERES observations. This part of the analysis needs a lot more thought before it is published.

2) A strong emphasis is placed upon the increased concentration of mass flux in the zonally averaged flow near the equator, an increase and narrowing in the upward branch of the Hadley Circulation (ref. 24). This arises from two reasons. The most important may be the tendency of

CMIP5 models to produce an enhanced warming in the tropical eastern Pacific and a consequent increase in precipitation near the equator in the East Pacific, not unlike a perennial El Niño state. The tropical precipitation does not increase uniformly, but changes shape to have a little less Walker circulation and more Hadley circulation. This is shown and discussed in the recent AR5 IPCC report in Chapters 12 and 14 (e.g. Fig. 12.10). The IPCC (IPCC, WGI, 2013) authors conclude that this aspect of the response of the AR5 models, the transition to a different ENSO state, remains uncertain, although the precipitation response seems quite robust across the models, since most produce more warming in the east Pacific than the west Pacific Ocean. Not all the models produce an El Niño-like response, however, and many do not produce a realistic ENSO cycle, so the IPCC authors remain doubtful.

The second reason this might occur was discussed by Held and Soden (2006), and many subsequent authors. As the moisture in the surface air increases exponentially with surface temperature, less convective mass flux is required to balance radiative cooling of the atmosphere, even though the latent energy available to drive convection increases. This tends to produce a greater increase in high precipitation rates than average precipitation. If high cloud fraction is proportional to mass flux and independent of surface temperature, then we should expect the high cloud fraction to decrease in a warmed Earth. Climate models appear to do this, and there is also a small decrease in tropospheric relative humidity associated with the reduction in high cloud area. This response of the climate models is highly uncertain, however, since climate models do not explicitly resolve the mesoscale circulations that maintain tropical high cloud associated with deep convection. Moreover, the answer to the question depends critically on the microphysics of ice in the upper tropical troposphere, which is also uncertain in climate models. It could be that the high cloud areas associated with convective cores increases in a warmed Earth along with the more intense updrafts that might be expected. These questions are yet to be satisfactorily answered. In cloud-resolving models with relatively simple microphysics the high cloud area does not seem to respond at all to surface temperature until pressure effects or the stratopause intervene. It would be nice to be able to constrain this issue with observations, but without a more careful analysis of what phenomena are causing the correlated changes in tropical mean high cloud and SST, one cannot accept the simple correlations used here as valid evidence.

The paper takes a great deal of effort to show the high cloud fraction and OLR are related, but this is well known.

Reviewer #2 (Remarks to the Author):

Review of 'Tightening of Hadley ascent key to radiative control on precipitation change under warmer climate' by Dr. Hui Su and co-authors.

In this manuscript the authors link statistically the tightening of the Hadley-cell ascending branch, via upper-level cloud fraction and their impact on atmospheric longwave cooling, to global mean precipitation change. The method is effectively one of these popular emergent constraints, wherein some observable measure is statistically related to a poorly observable quantity, such as climate sensitivity or in this case global mean precipitation change, using a climate model ensemble.

The first thing one needs to ask is whether the constraint is physically plausible (Klein and Hall, 2015)? I think that in this case the answer is yes, the atmospheric cooling is to a large extent controlled by longwave cooling from clear regions, and so a shrinkage of the areal extent of the cloudy regions should lead to more global mean precipitation.

My main problem with this work, however, is that the spread in hydrological sensitivity among models, when accounting for forcing adjustment and remnant warming, is much smaller than it might seem from common analyses (Andrews et al. 2009, Fläschner et al. 2016), and so the

statistical leverage of the CMIP5 ensemble to constrain Earth's hydrological sensitivity is weak. The factor of two spread (100 percent) found in the RCP8.5 experiment (Figure 4) is likely in part due to differences in forcing and temperature sensitivity, and the two referenced studies find spreads of only 50 percent. In any case this is much less than the widely touted 1-3 percent per Kelvin spread (200 percent) found in some misleading studies. For instance, differences in aerosol absorption, which has nothing to do with Hadley ascent tightening, among models running the scenario could impact global mean precipitation and thus create artificial correlations.

In my mind the authors would need to demonstrate that their emergent constraint method works on a clean case, such as the abrupt4xCO2 experiment whereby they need to separate out fast adjustments. If this works, which I doubt, then the required revisions are probably manageable within a normal revision cycle.

- - -

Minor comments:

21, I am not sure I would call the tightening the 'driver', rather the cause must be something else, e.g. convective precipitation efficiency.

32-33, The fact is that regional precipitation change is what matters for impacts and global mean precipitation is not related to this. The reasons for studying global mean precipitation are more academic, i.e. a manifestation of radiative feedbacks to climate change.

36, I would avoid citing these studies for the 200 percent uncertainty in hydrological sensitivity as they include forcing adjustment and non-equilibrium response.

126, I would avoid using the word 'strong' here.

138-140, I am not sure I understand the distinction between the fins and the iris-effect?

153, Here it might be worthwhile placing these cloud cover reductions in perspective with Lindzen's -22 %/K estimates.

163, Variations in OLR is further easier to measure and more directly linked to climate change feedback.

170, Also cloud emissivity is important.

218, Cherry-picking the model ensemble is not appreciated. Please provide the correlation across all models.

253, see my comment for line 21.

285-292, why use different scenarios?

353-354, this appears rather speculative.

356-358, If a systematic bias in the CloudSat data is introduced after a certain date, why not simply use the data up until this point?

Figure S5, please use the same scale in all panels.

Figure S6, please do not stretch panel n.

The supplement contains dead links to references.

- - -

Klein and Hall, Emergent constraints for cloud feedbacks. *Curr. Clim. Change Rep.* 1, 276-287 (2015).

Andrews et al. A surface energy perspective on climate change. *J. Clim.* 22, 2557-2570 (2009).

Fläschner et al. Understanding the intermodal spread in global-mean hydrological sensitivity. *J. Clim.*, 29, 801-817 (2016).

Reviewer #3 (Remarks to the Author):

Review of "Tightening of Hadley ascent key to radiative control on precipitation change under warmer climate" by H. Su et al.

The authors combine an array of state of the art satellite and climate model datasets to identify a potential emergent constraint on global precipitation change that relates to a narrowing of the Inter Tropical Convergence Zone (ITCZ) rainy belt through associated impacts on the outgoing longwave radiation. They argue that the mechanism involved links the thermodynamic and dynamic constraints on tropical precipitation change and is therefore a substantial advance to fundamental scientific understanding with notable applicability to future climate projections and related impacts on society. I have a number of mainly minor concerns and questions about the methods and the important conclusions drawn and as such I consider if the authors can address these comments then the manuscript will be suitable for publication in Nature Communications.

GENERAL COMMENTS

1) Given that interannual variability is dominated by ENSO with its associated pattern of circulation response, can links to long term climate change relate to a trend toward a more El Nino-like state shown by many models (e.g. Cai et al. 2014 *Nature Clim.* doi:10.1038/nclimate2100)? If so, the link depends upon the robustness of the model dynamical response (which may in any case link to fundamental thermodynamic responses of mass flux and gross moist stability as argued in the current study).

2) How much of the zonal mean response is a "tightening" of the Hadley Cell and how much is a reorganisation of the ITCZ into a more zonal configuration?

3) Is there any link to model response in "tightening" and the present day biases in ITCZ location and associated biases in cross equatorial energy transports (e.g. Loeb et al. 2016 *Clim. Dyn.* doi:10.1007/s00382-015-2766-z)?

4) If some of the South Pacific Convergence Zone (SPCZ) strays outside the 20S-20N zone this could also influence the amount of the ascending branch of the Hadley circulation that is sampled. Are similar results found for 30S-30N?

5) Can tightening be diagnosed by considering the probability distribution of vertical motion or by considering changes as a function of surface temperature relative to tropical mean (ie in regime rather than geographical space)?

6) Satellite infrared retrievals sample clear-sky outgoing longwave for cloud-free regions which are drier than cloudy regions and may have stronger sensitivity to temperature changes compared with the models. This of course also affects cloud radiative effect although the influence may be

small. Did the authors check this using model data to mimic satellite sampling or do past studies show this effect is small?

7) Figure 1a/b is rather complicated! I think at the very least a legend would be helpful. It may also be useful to note in the text that negative omega signifies ascent.

8) The work of Wodzicki and Rapp (2016) JGR doi:10.1002/2015JD024458 appears to be relevant

9) Although differences in surface net longwave do not seem to explain the model diversity, it may be useful to emphasise that longwave radiative cooling is strongly enhanced through reduced net upward longwave flux at the surface as the tropics warm

Minor Comments/Corrections

L20 - Abstract: "under warmer"  "in a warmer"

L41 - LvP is an approximation since snowfall involves the latent heat of fusion also (e.g. Loeb et al. 2016 Clim. Dyn.)

L47/48 - this is slightly vague: global mean precipitation and its variability are determined to a large extent by longwave radiative cooling (LWC) but maybe "primarily constrained" is not quite the best phrase?

L53 - "in controlling"

L57 - it should probably be stated that global mean is being discussed?

L68 - "a human's eye"

L140 - presumably increased cloud fraction (CF) is linked to higher upper tropospheric humidity as well as the size of the radiator fin

L143 - this seems inaccurate: in the tropics very little surface emission escapes to space: I suggest "lower tropospheric thermal emission" is more precise

L157 - given the large spatial reorganisation of circulation systems during ENSO, the short term "tightening" could also involve a more zonally uniform ITCZ as well as a physical tightening.

L164 - the period of CERES data should be stated, presumably 2000-2015? Would a large El Nino like 1997/8 or 2015/16 influence this correlation. A longer model record or combination of CERES/ERBS data may help to test this (e.g. Allan et al. 2014 GRL doi:10.1002/2014GL060962) although I expect this sensitivity is quite robust.

L168 - "that most cloud"

L172 - "when the surface"

L177 - strengthened descent may be more accurate

L179 - "is opposed to"  "is the opposite of"?

L201 - can the reverse also be argued e.g. increased latent heating leads to greater longwave thermal emission?

L223 - some further clarification of the "bias removal procedure" may be appropriate here

L230 - I find it difficult to distinguish cyan from blue

L237 - "effective venue"  "effective step/strategy/method"?

L240 - do models with unrealistic shortwave absorption sensitivity to temperature (e.g. GISS models) also display unrealistic longwave sensitivities?

L249 - "under warmer"  "under a warming"

L260 - "multi"  "multiple"

L267-270 - I was slightly confused here: are values taken from other studies here or are they calculated from the models used in the present study for RCP8.5?

L275 - "promises"  "promise"

L279, L309 - "same as"  "consistent with"

L279 - The 1995-2005 period overlaps with less than 5 years of CERES unless a combined ERBS record is used (although clear-sky would probably have to be approximated from reanalysis).

L289 - a reference to fast and slow precipitation responses should be included here.

L315, L341 - a precise time period would be useful (also for the other datasets rather than having to check supplementary information).

Fig. 4 - can the short term $LvdP/dTs$ from observations also be shown or is this too uncertain (see further comments below)

L593 - "a-axis"  "x-axis"?

Supplementary:

L28 - it could be noted that this is close to that expected from the Clausius Clapeyron equation at cold upper troposphere temperatures (e.g. about 14%/K at 200K). How does the AIRS/MLS estimate compare with other UTH data (e.g. Soden et al. 2005 Science, doi:10.1126/science.1115602). Two decimal places is not necessary (e.g. 9.2-15.6%/K is fine).

L37, L51-54, L65, L68, L71 - there are errors in the PDF I downloaded ("Error! Reference Source not found").

L52, L66 - "recipitation"  "precipitation"

L46+ - when considering long term $dOLR/dTs$ the influence of greenhouse gas increases is important e.g. $(dOLR/dGHG) \times (dGHG/dTs)$: does this need to be accounted for?

L54 - the CMAP trends have previously been found to be unreliable due to calibration on atoll data with real trends so I do not think it is appropriate to even consider this dataset. Using the GPCP and TRMM (1997+) data, although perhaps not independent, surely provides a more robust representation and is probably closer to the $dOLR/dTs$ relationship I think (it would be interesting to compare with CERES OLR for 2000-2015). I found global dP/dTs of 2.8%/K and a relationship between GPCP global P and ERA interim total atmospheric cooling quite close to unity (Allan et al. 2014 Surv. Geophys. DOI 10.1007/s10712-012-9213-z) for 1988-2008.

L65 - "same as"  "consistent with"

L69 - "orrelation"  "correlation"

L75 - I am not completely convinced by this method. Thinking about model "o" for example in Fig. 4, the LvdP/dTs seems to be "over-corrected" due to its deviation from the fit line. A little clarification may just help here.

L86 - is there explanation for the lack of relation between dCF/dTs and ECS? High clouds have a small net radiative effect but I guess also that there is a diverse mix of responses for different meteorological regimes and interannual variations may not be a good proxy for long term cloud changes?

Fig. S6 - panel n should be regular size to help comparison and an x-axis should be added to panel l.

Overview of Response to Reviewers:

We thank the reviewers for detailed comments and helpful suggestions. In the revision, we have added 8 RCP4.5-historical pairs of model simulations so that the total number of long-term simulations is 23 (21 in some analyses), which is the same as that of the AMIP5 runs, to increase statistical robustness. We have also conducted additional analysis following the reviewers' suggestions, including the use of temperature-mediated precipitation change per unit surface warming from the "abrupt4xCO2" experiments as a measure of hydrological sensitivity for all available models, and a new approach to apply the observational emergent constraints on the hydrological sensitivity. The new analysis results corroborate our original conclusions. We have addressed the three reviewers' comments point-by-point with supporting evidence. The manuscript and supplementary materials have been modified extensively, including reordering several sections and figures and adding new figures.

We have carefully addressed Reviewer 1's concerns about the cause-and-effects and the validity of emergent constraints on future climate change. In the revision, we have carefully addressed the two-way coupled interactions between circulation, clouds and surface temperature instead of simple one-way relations. The new approach to apply the emergent constraints highlights the usefulness of the short-term observations in identifying the *likely* models (at the 95% confidence level). We do not simply use the value of interannual sensitivity to replace that of long-term sensitivity.

We respectfully disagree with Reviewer 1's assessment that "*the model analysis is not new, but has been described before*" because "*the paper takes a great deal of effort to show the high-cloud fraction and OLR are related, but this is well known.*" The assessment is not correct because the high-cloud fraction and OLR relation is just a stepping stone towards the conclusion about the role of large-scale circulation in the interactions between high-cloud fraction (CF), surface temperature and the radiative control of global-mean precipitation. What we present in the paper is that the inter-model spread in $dOLR/dT_s$ is primarily caused by the model disagreement in dCF/dT_s , which is physically plausible but has not been demonstrated before. Neither do we agree with Reviewer 1's claim that our "*estimate of $dOLR/dT_s$, $3.79 W m^{-2} K^{-1}$, is outside decades of research*". As elaborated in the Response to Reviewer 1, this estimate is consistent with previous studies.

We think the results presented in the manuscript are robust and novel. They provide compelling evidence for the important role of the *Tightening of Hadley Ascent* in governing the rate of global-mean precipitation change in response to increasing

greenhouse gases and the likely low biases of current climate model predictions of hydrological sensitivity. In addition, the results have profound implications for regional precipitation change and thus direct societal impacts because the tightening of ascending regions is associated with the intensification of precipitation in heavily precipitating regions, i.e., the “wet get wetter” response to global warming. Our study is a timely contribution to climate science that would attract broad interest.

Comments from Reviewers are highlighted in red and italicized, followed by our responses in blue. All changes are marked up in the revised manuscript in orange color. A final clean version is submitted online as well.

Response to Reviewer 1

We appreciate Reviewer 1’s critical assessment. We have carefully considered his/her comments and provided our explanations with supplementary analysis results.

“This paper combines two aspects. One is an observational analysis that shows a negative correlation between high-cloud cover and sea surface temperature in the tropics from 20N-20S, and a strikingly large positive correlation between OLR and surface temperature. Another is an analysis of models that compares the sensitivity of high-cloud fraction and outgoing longwave radiation to surface temperature. By combining these two things the authors imply that the reduction of high-cloud fraction with warming increases the cooling of the atmosphere and thereby controls the rate at which global mean precipitation increases with surface temperature. By comparing the observed covariance between cloud fraction and surface temperature in observed month-to-month variability with model results, the authors conclude that the sensitivity of precipitation to temperature should be greater in models than it is currently represented to be.

I have two primary objections to the research. First: the applying the observational analysis to climate sensitivity is inappropriate, so that the presentation of this analysis as an emergent constraint is not yet warranted. It is very naive. The values of some covariances are outside the range expected from decades of research, so their interpretation as a climate sensitivity constraint is highly questionable. Second: the model analysis is not new, but has been described before.

1) In the model analysis the authors correlate the deviations of monthly mean cloud fraction anomalies and OLR anomalies with monthly surface temperature anomalies. It is not demonstrated what produces these anomalies and whether they represent a response of clouds to SST or a response of SST to clouds. Therefore to characterize them as useful in a feedback constraint context is not warranted without much further research. One could hypothesize that these changes are associated with natural variability of the amount of tropical deep convection averaged over the tropics. An example of this is the Madden Julian Oscillation which produces coordinated changes in high-cloud fraction and SST over a large enough area to affect the whole tropics. This would occur in both summer and winter. When the SST is high clouds are more likely to form. When they form they cool the surface through reflection of solar radiation and through enhanced vertical

transport of energy. Thus SST and cloud fraction can be negatively correlated. If this is the cause of the month-to-month variability, then it is not measuring the response of clouds to SST in a climate feedback sense, but rather a combination of clouds forcing SST and SST forcing clouds in disequilibrium oscillations. This might be expected to produce a larger negative correlation between cloud fraction and SST than one might get in response to global climate change. Therefore, I don't think it should be used to gauge the reliability of model simulations of climate change, and should not be used as a constraint, until it can be more deeply studied and shown to be a good analog for the equilibrium response to surface warming.”

First, we would like to point out that the correlations on the interannual and centennial time scales we present in the manuscript do not represent the cloud-SST covariances on the MJO time scale, because the interannual sensitivities are based on the regression slopes of 5-month running means of high-cloud fraction anomalies onto T_s anomalies so that the 30-90 day MJO contributions are negligible. The centennial rates are based on the 100-year differences between the two 25-year means from the 21st and 20th centuries. However, there are similarities between the cloud-SST relations on all these time scales, as we have shown in the paper and further elaborated below. The similarities on different time scales are the basis for inferring long-term changes from short-term variabilities.

Figure 1. Spatial maps of the regression coefficients of AMIP5 model simulated high-cloud fraction (color shadings, in %/K) and surface temperature (contours, in K/K) onto the tropical-mean (20°S-20°N) surface temperature from 1995 to 2005 for 14 model simulations and the multi-model-mean (the bottom right panel). **The Central and Eastern Pacific SST are warmer than other oceans due to El Niño.**

On the interannual time scales, the tropical-mean surface temperature variations closely follow the El Niño SST anomalies (Chiang and Sobel 2002; Su et al. 2001; 2003), which are driven by atmosphere-ocean coupling (Neelin et al. 1998). Figure 1 above shows the positive regressions of local T_s onto the tropical-mean T_s over most of the tropical oceans except for the Western Pacific, and the largest regressed values occur in the eastern and Central Pacific. High-cloud fraction increases over the Eastern and Central Pacific where warm SST anomalies are located, while high-cloud fraction decreases over the Western Pacific and to the north and south of the warm SST anomalies, as reported by previous studies (Ramanathan and Collins 1991; Zelinka and Hartmann 2011; Su and Jiang 2013). In the models, the high-cloud responses to the El Niño SST anomalies are approximately captured, but with differences in magnitudes and locations (color shadings in Figure 1). When we examine the co-variance of high-cloud fraction and surface temperature in each grid box, positive correlations are overwhelming over ocean (except for the small areas over the northeast Pacific and isolated spots in the Indian Ocean), while the correlations are predominantly negative over land (Figure 2). The strong positive correlation between high-cloud fraction and local SST anomalies over the oceans suggests that high-cloud fraction primarily responds to the local SST warming on the interannual time scale. Over the limited oceanic areas of negative correlations, anomalous descent can reduce high-cloud fraction even if underlying SST anomalies are positive because the relative warmth of local SST to tropical-mean SST is important for the circulation anomalies. Over tropical land, surface temperature responds fast to cloud variations driven by circulation changes, resulting in negative correlations (reduced clouds during El Niño contribute to surface warming) (Neelin and Su 2005).

When we consider the **tropical-mean** high-cloud fraction and surface temperature anomalies, a negative correlation is found (Figure 3a and Supplementary Figure 5 in the manuscript). This is true even when we include only tropical oceans in the tropical-mean. The sharp contrast between the spatial patterns in Figure 1 and Figure 2 suggests that the tropical-mean high-cloud-SST relation is not simply the averaging of local relations in which SST forces clouds and clouds feedback onto SST. The role of circulation change must be considered. The tightening of tropical ascents represents the overall effect of the changes in the Hadley Circulation and the Walker Circulation on the tropical-mean. The narrowing of ascending regions with El Niño warming would produce smaller high-cloud fraction. The reduced high-cloud amount is associated with less cloud longwave warming in the atmosphere, which can further promote the tightening of ascents. The decrease of tropical-mean high-cloud fraction could further produce a net warming effect on the tropical-mean surface temperature because the cloud shortwave effect overcomes its longwave effect, providing a positive feedback (Zelinka and Hartmann 2011). Hence, we agree with the reviewer that the negative correlation between the tropical-mean cloud fraction and T_s on the interannual time scale includes the two-way interactions between cloud response to surface warming and subsequent cloud feedback onto SST. However, we emphasize that the model disagreement on the extent of the tightening is an important driver of the model differences in tropical-mean high-cloud amount reduction and the longwave radiative feedback, and eventually the global-mean precipitation change in response to surface warming.

Figure 2. Spatial maps of the regression coefficients of AMIP5 model simulated high-cloud fraction (color shadings, in %/K) onto the local surface temperature for 14 model simulations and the multi-model-mean from 1995 to 2005.

On the centennial time scale, climate models tend to produce an “El Niño like” SST warming pattern, a fact pointed out by Reviewer 1 and discussed in the IPCC AR5 report and many previous studies, even though it is not clear whether this tendency is realistic. Because of the preferred SST warming pattern, the projected high-cloud fraction changes (normalized by the tropical-mean T_s change) exhibit increases over the relatively warmer equatorial Central and Eastern Pacific and decreases over the Western Pacific and outside the relatively warmer regions (Figure 3), which is approximately similar to the El Niño cloud response in Figure 1, albeit with differences in certain regions. On the tropical-mean, most models simulate a reduction of the high-cloud fraction compared to the 20th century climatology, because of the tightening of Hadley Ascent (Figure 3b in the manuscript). The decrease of high-cloud amount could provide further warming to the SST because the reduced shortwave cooling effect overcomes the reduced longwave warming effect. The two-way interaction of SST forcing the cloud and cloud feedback to SST under global warming is analogous to those on the interannual time scale, despite the fact that the amplitudes of changes on the two time scales are very different (Figure 1 and Figure 3). The resemblance between the spatial patterns shown in Figure 1 and 3 and those tropical-mean relationships presented in the manuscript provide the physical basis for linking the cloud variabilities on the short-term and the long-term, i.e., the circulation-cloud-radiation-precipitation interactions operate similarly on the two time scales.

Figure 3. Spatial maps of the changes of CMIP5 high-cloud fraction (color shadings, in %/K) and surface temperature (contours, in K/K) normalized by tropical-mean (20°S-20°N) surface temperature change using 12 RCP4.5-historical model simulations (CNRM_cm5 and INM_cm4 do not have cloud fraction profile outputs). The changes are the differences between the 25-year averages in 2074-2098 and 1980-2004. **The Central and Eastern Pacific SST are warmer than other oceans due to the simulated “El Niño like” warming pattern in climate models.**

In summary, we agree with the reviewer that the SST-cloud correlations do not simply represent the cloud response to SST. We have modified the manuscript to highlight the two-way interactions between SST and clouds, as well as for circulation, clouds and radiation relations. As the purpose of this study is to highlight the role of tropical circulation in the cloud-radiation-precipitation interactions and provide observational constraints on the relationships, we think it is valid to use the simple correlations to represent the coupled relations in a compact way. We have added more descriptions in the manuscript to explain the underlying interactive physical processes.

We note that when we apply interannual observations as “emergent constraints” to confine the range of future precipitation change predictions, we do not simply equalize future sensitivities with present-day short-term counterparts. As Reviewer 1 pointed out, the short-term and long-term sensitivities are not of the same magnitude. For example, the magnitudes of centennial dCF/dT_s are generally smaller than those of interannual dCF/dT_s (Figures 1 and 3 here; Figure 2 in the manuscript x-axis). Following previous studies that populated the “emergent constraint” approach (e.g., Fasullo and Trenberth 2012; Sherwood et al. 2014), the procedures of applying the “emergent constraint” are 1) to find the quantities (in our case, $dOLR/dT_s$ and dP/dT_s) for which the present-day inter-

model spread is significantly correlated with the inter-model spread in future predictions; 2) to use the observations to identify the models that produce realistic present-day simulations for those relevant quantities; 3) to identify the **likely** future predictions based on the “better-performing” models. This approach is equivalent to applying weights (1 for “better” and 0 for “worse” models) to the original entire model ensembles based on present-day model performances. This approach is well established in recent literature as long as the metrics are physically based (Klein and Hall 2015). In our case, the physical basis is valid, and supported by Reviewer 2 and Reviewer 3.

“The sensitivity of OLR to surface temperature in line 164 is about twice the value obtained in numerous observational and theoretical studies. This suggests that the change of SST driven by the cloud cover is contributing to this value. The simple, nearly blind comparison of monthly, tropical mean anomalies of OLR, high-cloud, and SST done in this paper is not adequate to overturn decades of research. Most models do not simulate the MJO properly, and that might be a reason why the short-term variability in those models cannot produce the very large value inferred from the CERES observations. This part of the analysis needs a lot more thought before it is published.”

Our estimate of the interannual sensitivity of OLR to surface temperature averaged over 20°S-20°N in original Line 164, is $3.79 \text{ W m}^{-2} \text{ K}^{-1}$. This value is consistent with a number of existing studies. We do not know the basis for the reviewer’s claim that this number is outside the range of decades of research. The following studies contradict the reviewer’s position.

Mauritsen and Stevens (2015, Nature Geoscience, their Figure 2, Table S2) showed that $d\text{OLR}/dT_s$ based on the CERES TOA fluxes (20°S-20°N) from 2000 to 2013 is $4.05 \pm 0.82 \text{ W m}^{-2} \text{ K}^{-1}$. The slight difference from our result is because we use a longer record of CERES data than theirs, 3/2000 to 10/2015, and we applied 5-month running averaging and they used de-trended data. We were able to reproduce their result exactly if the same data period and de-trending were used.

Lindzen and Choi (2009) reported that the tropical $d\text{OLR}/dT_s$ using ERBE data from 1985-1999 for selected SST perturbations is $3.5 \pm 0.82 \text{ W m}^{-2} \text{ K}^{-1}$. Trenberth et al. (2010) compared various ways to select the data and with/without the Pinatubo effect and produced $d\text{OLR}/dT_s$ between 2.7 and $3.3 \pm 0.5 \text{ W m}^{-2} \text{ K}^{-1}$. Lindzen and Choi (2011) showed that the tropical $d\text{OLR}/dT_s$ is close to $5.0 \pm 1.3 \text{ W m}^{-2} \text{ K}^{-1}$ for the combined ERBE (1985-1999) and CERES (2000-2008) period. Foster and Gregory (2006) reported that the $d\text{OLR}/dT_s$ for 1985-96 is about $4.0 \text{ W m}^{-2} \text{ K}^{-1}$ based on the ERBE data. We analyzed the ERBE data for 1985-1999 and also combined the ERBE data with CERES to obtain the $d\text{OLR}/dT_s$ for the 1995-2005 period (same for the AMIP5 model simulations in the paper) and various sub-periods. All the analyses found that the tropical $d\text{OLR}/dT_s$ is close to $4 \text{ W m}^{-2} \text{ K}^{-1}$ consistently.

We have examined the components of the $d\text{OLR}/dT_s$, i.e., the clear-sky $d\text{OLR}_{\text{clr}}/dT_s$ and the cloudy-sky $d\text{CRE}_{\text{lw}}/dT_s$, and found consistent results with previous studies such as Allan (2006) (their Table 3) and Chung et al. (2010) (their Figure 2). We have also

analyzed the TOA OLR from AIRS and ISCCP and obtained similar values. All the data used in the study will be made available to the public when the paper is published.

It is worth pointing out that the value of $3.79 \text{ W m}^{-2} \text{ K}^{-1}$ for $d\text{OLR}/dT_s$ is averaged over the tropical 20°S - 20°N because of our focus on tropical high-clouds. When the global average is considered, the $d\text{OLR}/dT_s$ from the CERES data (3/2000-10/2015) is $2.11 \pm 0.43 \text{ W m}^{-2} \text{ K}^{-1}$. The tropical and global difference is similar for the ERBE data from 1985 to 1999. Murthy et al. (2009) (their Table 1) provides a comprehensive summary of preceding studies. The global-mean $d\text{OLR}/dT_s$ ranges from 1.64 to $3.8 \text{ W m}^{-2} \text{ K}^{-1}$, with most of the estimates around 2.1-2.5 $\text{W m}^{-2} \text{ K}^{-1}$, consistent with our estimate using current CERES EBAF data. We believe that Reviewer 1 might have confused the tropical sensitivity with the global one. We have clarified the tropical and global difference in the revised version.

In CMIP5, most models underestimate the sensitivity of OLR to surface warming, a fact pointed out by previous studies (Lindzen and Choi 2011; Mauritsen and Stevens 2015), and emphasized in this study. We have extended the previous studies to address the role of large-scale circulation in this low bias in $d\text{OLR}/dT_s$ across the models. Our analysis suggests that the models' biases in capturing the tightening under warming conditions could be a reason for the low bias in $d\text{OLR}/dT_s$. We don't disagree with Reviewer 1 that the model deficiency in simulating MJO may contribute to the bias in the longwave sensitivity. However, it is beyond the scope of this study to address all possible model deficiencies. We recognize further studies are needed to identify the key model physical parameters responsible for the biases in simulating the circulation, cloud and precipitation changes. We have added this point in the discussions.

“2) A strong emphasis is placed upon the increased concentration of mass flux in the zonally averaged flow near the equator, an increase and narrowing in the upward branch of the Hadley Circulation (ref. 24). This arises from two reasons. The most important may be the tendency of CMIP5 models to produce an enhanced warming in the tropical Eastern Pacific and a consequent increase in precipitation near the equator in the East Pacific, not unlike a perennial El Niño state. The tropical precipitation does not increase uniformly, but changes shape to have a little less Walker circulation and more Hadley circulation. This is shown and discussed in the recent AR5 IPCC report in Chapters 12 and 14 (e.g. Fig. 12.10). The IPCC (IPCC, WGI, 2013) authors conclude that this aspect of the response of the AR5 models, the transition to a different ENSO state, remains uncertain, although the precipitation response seems quite robust across the models, since most produce more warming in the east Pacific than the west Pacific Ocean. Not all the models produce an El Niño-like response, however, and many do not produce a realistic ENSO cycle, so the IPCC authors remain doubtful. The second reason this might occur was discussed by Held and Soden (2006), and many subsequent authors. As the moisture in the surface air increases exponentially with surface temperature, less convective mass flux is required to balance radiative cooling of the atmosphere, even though the latent energy available to drive convection increases. This tends to produce a greater increase in high precipitation rates than average precipitation. If high-cloud fraction is proportional to mass flux and independent of surface temperature, then we

should expect the high-cloud fraction to decrease in a warmed Earth. Climate models appear to do this, and there is also a small decrease in tropospheric relative humidity associated with the reduction in high-cloud area. This response of the climate models is highly uncertain, however, since climate models do not explicitly resolve the mesoscale circulations that maintain tropical high-cloud associated with deep convection. Moreover, the answer to the question depends critically on the microphysics of ice in the upper tropical troposphere, which is also uncertain in climate models. It could be that the high-cloud areas associated with convective cores increases in a warmed Earth along with the more intense updrafts that might be expected. These questions are yet to be satisfactorily answered. In cloud-resolving models with relatively simple microphysics the high-cloud area does not seem to respond at all to surface temperature until pressure effects or the stratopause intervene. It would be nice to be able to constrain this issue with observations, but without a more careful analysis of what phenomena are causing the correlated changes in tropical mean high-cloud and SST, one cannot accept the simple correlations used here as valid evidence.”

We appreciate the reviewer for sharing with us his/her insights on the possible mechanisms for the tightening of Hadley Circulation and the reduction of high-cloud fraction under global warming.

- 1) The SST warming pattern: we agree with the reviewer that climate models tend to produce an “El Niño” SST warming pattern and whether this is realistic remains to be proved. The differences in the simulated SST warming patterns do have strong influence on the circulation, cloud and precipitation changes. Please see our response to Reviewer 3 (#1 in General Comments) and Figure 4 there. Note that the SST warming patterns result from the interactions between circulation and cloud feedbacks. Again, the long-term relationships have strong analogy to the interannual variabilities.
- 2) The simple thermodynamic argument by Held and Soden (2006) is useful in explaining the circulation, cloud and precipitation changes. In spite of large uncertainties in model physics, climate models tend to capture the weakening of large-scale circulation, but with varying magnitudes. Our study addresses the importance of the tightening from the perspective of inter-model spreads. This result has sound physical basis but has not been shown before. Furthermore, the narrowing and strengthening of the Hadley ascent have been largely ignored in existing literature, which focuses on the weakening of large-scale circulation under global warming.
- 3) It is true that ice microphysics is highly uncertain in climate models. It could be a large source for the model bias in the high-cloud fraction and the longwave radiative feedback. Our analysis targets the tightening of ascents and reduction of high-cloud fraction. We find that the tightening biases could explain about 42% ($R=0.65$) of the variance of the across-model differences in dCF/dT_s . This implies that other processes that the reviewer indicated (the SST warming pattern, mesoscale circulation, ice microphysics and the processes important for the MJO simulations) can also make sizeable contributions. Our study identifies one

important factor but does not exclude other possible factors. We have modified the text to state the importance of other processes not analyzed in this study.

A recent paper by Bony et al. (2016, PNAS) suggested the increase of upper-tropospheric stability associated with surface warming would reduce the clear-sky radiatively driven mass convergence and thus the height and amount of upper tropospheric anvil clouds. The decrease of upper level high-cloud amount would make the cloud radiative heating increasingly localized and further promotes convective aggregation and the narrowing of moist convective areas, forming a positive feedback loop. The decrease and narrowing of ascending areas have been found in several idealized GCM and CRM simulations. Our study supports the linkage between the tightening of tropical ascending regions and the tropical-wide iris effect from the angle of the inter-model spreads, and relates to the circulation/high-cloud changes with global-mean precipitation change. The state-of-the-art observations are then used to constrain the hydrological sensitivity. Therefore, this study is innovative and has its own merits. In the revision, we have expanded the discussions of various model uncertainties and stressed the coupled relationships between the circulation, clouds, surface temperature, radiation and precipitation.

“The paper takes a great deal of effort to show the high-cloud fraction and OLR are related, but this is well known.”

Yes, it is well known that OLR depends on cloud fraction. *What we show in the paper is that the inter-model spread in OLR change is predominantly dependent on the spread in the cloud fraction change. Although this is not a surprising result, it is not a guaranteed result, because OLR also depends on cloud top height, cloud thickness and atmospheric water vapor.* The model differences in the $dOLR/dT_s$ could be affected by the simulation differences in the factors other than high-cloud fraction. More importantly, the OLR-cloud fraction relation is only a part of our analyses, as a stepping-stone in the attribution of error sources. *The most significant and novel aspect of our study is the linkage between the tightening of Hadley Ascent and high-cloud change and their radiative control of the hydrological sensitivity.* The contraction of equatorial ascent has not been discussed extensively in the literature, compared to the poleward expanding of the Hadley descent zone. No studies examined the connection between the tightening of Hadley ascent and the global precipitation sensitivity. Our study presents robust evidence of the importance of the tightening of Hadley Ascent to the longwave radiative cooling and the intensification of the hydrological cycle. The strong analogy between the interannual and centennial circulation-cloud-radiation-precipitation relationships has never been shown before. No publications have provided an observational constraint on the hydrological sensitivity based on the longwave radiative feedback.

It is not fair to disparage the novelty of this study based on a small step in the analysis, i.e., the relation between the high-cloud fraction and OLR. In the revision, we have greatly shortened the section on the OLR and cloud fraction relations and re-arranged the figures to emphasize the new findings.

Response to Reviewer 2

We thank Reviewer 2's thoughtful comments and suggestions to sharpen the results. We have replaced the old Figure 4 with the new analysis results of applying the observational constraints on the temperature mediated precipitation change from the "abrupt4xCO₂" experiments. A detailed point-by-point response is provided below.

Major Comments:

Review of 'Tightening of Hadley ascent key to radiative control on precipitation change under warmer climate' by Dr. Hui Su and co-authors.

In this manuscript the authors link statistically the tightening of the Hadley-cell ascending branch, via upper-level cloud fraction and their impact on atmospheric longwave cooling, to global mean precipitation change. The method is effectively one of these popular emergent constraints, wherein some observable measure is statistically related to a poorly observable quantity, such as climate sensitivity or in this case global mean precipitation change, using a climate model ensemble.

The first thing one needs to ask is whether the constraint is physically plausible (Klein and Hall, 2015)? I think that in this case the answer is yes, the atmospheric cooling is to a large extent controlled by longwave cooling from clear regions, and so a shrinkage of the areal extent of the cloudy regions should lead to more global mean precipitation.

My main problem with this work, however, is that the spread in hydrological sensitivity among models, when accounting for forcing adjustment and remnant warming, is much smaller than it might seem from common analyses (Andrews et al. 2009, Fläschner et al. 2016), and so the statistical leverage of the CMIP5 ensemble to constrain Earth's hydrological sensitivity is weak. The factor of two spread (100 percent) found in the RCP8.5 experiment (Figure 4) is likely in part due to differences in forcing and temperature sensitivity, and the two referenced studies find spreads of only 50 percent. In any case this is much less than the widely touted 1-3 percent per Kelvin spread (200 percent) found in some misleading studies. For instance, differences in aerosol absorption, which has nothing to do with Hadley ascent tightening, among models running the scenario could impact global mean precipitation and thus create artificial correlations.

In my mind the authors would need to demonstrate that their emergent constraint method works on a clean case, such as the abrupt4xCO₂ experiment whereby they need to separate out fast adjustments. If this works, which I doubt, then the required revisions are probably manageable within a normal revision cycle.

We thank the Reviewer's positive comments on the physical basis of the emergent constraint on the hydrological sensitivity. We agree with the reviewer that the model differences in the total precipitation change under the RCP8.5 scenario include the contributions of different radiative forcings and the differences in the fast response to direct radiative forcing in addition to the differences in the slow temperature-mediated

precipitation responses. The latter is a better measure of the hydrological sensitivity to surface warming, which has an inter-model spread about 1.5 instead of a factor of 3 mentioned in many studies that did not distinguish the fast and slow responses and the differences in the radiative forcings. We have modified the manuscript accordingly.

While applying the observations to constrain the temperature-mediated hydrological sensitivity, we found that the interannual $dOLR/dT_s$ is less correlated with the temperature-mediated precipitation response than the interannual dP/dT_s ($R=0.18$ and 0.64 respectively). To ensure robust statistics, we use the interannual dP/dT_s as the new “emergent constraint”. The range of the likely observational estimate (at the 95% confidence level) of dP/dT_s is determined by the combined knowledge of the direct measurements from the GPCP data during the period of 1995-2005 and the observed $dOLR/dT_s$ from CERES/HadCRU4 coupled with the modeled relationship between the interannual $dOLR/dT_s$ and dP/dT_s (see the revised section “Constraining hydrological sensitivity using observations”). Therefore, we have effectively utilized both CERES and GPCP data to obtain the best estimate of the observation-based interannual dP/dT_s . We have discarded the bias-removal procedure used in the original manuscript; instead, a new approach is implemented to identify the likely models based on the best estimate of the observation-based interannual dP/dT_s . We find that the five models that fit into the observed range of the interannual dP/dT_s all have hydrological sensitivity greater than the multi-model-mean. The across-model standard deviation for the hydrological sensitivity is reduced by 66% compared to the entire model ensemble.

We have replaced the old Figure 4 with the new results and modified text accordingly.

Minor Comments:

21, I am not sure I would call the tightening the 'driver', rather the cause must be something else, e.g. convective precipitation efficiency.

We have replaced the driver to “a key process” here and similar places in the text.

32-33, The fact is that regional precipitation change is what matters for impacts and global mean precipitation is not related to this. The reasons for studying global mean precipitation are more academic, i.e. a manifestation of radiative feedbacks to climate change.

We agree with the reviewer and added “regional” in the sentence.

36, I would avoid citing these studies for the 200 percent uncertainty in hydrological sensitivity as they include forcing adjustment and non-equilibrium response.

We have discussed the total precipitation change, fast and slow responses and changed the references here.

126, I would avoid using the word 'strong' here.

The word “strong” is removed.

138-140, I am not sure I understand the distinction between the fins and the iris-effect?

The radiator fins and iris-effects are closely related. Our study emphasizes the reduction of high-cloud fraction with surface warming is linked to the expansion of radiator fins and the increase of atmospheric longwave cooling rate. Therefore, in our study, the fins and the iris-effect are interchangeable, although the original “iris-effect” discussed in the literature can include both the longwave and shortwave cloud effects. We focus on the longwave effect only. We have clarified this in the manuscript.

153, Here it might be worthwhile placing these cloud cover reductions in perspective with Lindzen's -22 %/K estimates.

The reference to Lindzen’s -22%/K is added in the text.

163, Variations in OLR is further easier to measure and more directly linked to climate change feedback.

We have added this point in the text. Thanks for the suggestion.

170, Also cloud emissivity is important.

Added “cloud emissivity”.

218, Cherry-picking the model ensemble is not appreciated. Please provide the correlation across all models.

We have added 8 RCP4.5 models to the entire analysis of the long-term sensitivities and have used all available models for correlations throughout the manuscript unless there are missing data.

253, see my comment for line 21.

Replaced the word “driver” with “a key process”.

285-292, why use different scenarios?

We have discarded the results using the RCP8.5 scenarios; instead, we have examined the temperature-mediated precipitation change per unit warming from the “abrupt4xCO2” experiments. The original Figure 4 is replaced with the new analysis and the corresponding text is revised.

353-354, this appears rather speculative.

The speculation is removed.

356-358, If a systematic bias in the CloudSat data is introduced after a certain date, why not simply use the data up until this point?

We have used respective mean seasonal cycles before and after 2011 to compute the de-seasonalized anomalies for the period from 6/2006 to 6/2015. The large range of the tropical-mean high-cloud fraction anomalies from CloudSat/CALIPSO compared to other datasets could be due to the sparse spatial sampling of the active sensors. Using the period before 2011 results in nearly neutral regression slope and even larger uncertainties (not shown in the figure). We have added the caveats of the CloudSat/CALIPSO data in the text.

Figure S5, please use the same scale in all panels.

See the reply above. We cannot use the same y-axis scale for the CloudSat/CALIPSO data as the others because of the different ranges of the data. The x-axes are the same. This figure is now Supplementary 10.

Figure S6, please do not stretch panel n.

Done. This figure is now Supplementary 11.

The supplement contains dead links to references.

All links are corrected.

Response to Reviewer 3

We thank the reviewer's thoughtful comments and a point-by-point response is provided below.

Review of "Tightening of Hadley ascent key to radiative control on precipitation change under warmer climate" by H. Su et al.

The authors combine an array of state of the art satellite and climate model datasets to identify a potential emergent constraint on global precipitation change that relates to a narrowing of the Inter Tropical Convergence Zone (ITCZ) rainy belt through associated impacts on the outgoing longwave radiation. They argue that the mechanism involved links the thermodynamic and dynamic constraints on tropical precipitation change and is therefore a substantial advance to fundamental scientific understanding with notable applicability to future climate projections and related impacts on society. I have a number of mainly minor concerns and questions about the methods and the important conclusions drawn and as such I consider if the authors can address these comments then the manuscript will be suitable for publication in Nature Communications.

GENERAL COMMENTS

1) Given that interannual variability is dominated by ENSO with its associated pattern of circulation response, can links to long term climate change relate to a trend toward a more El Nino-like state shown by many models (e.g. Cai et al. 2014 Nature Clim. doi:10.1038/nclimate2100)? If so, the link depends upon the robustness of the model dynamical response (which may in any case link to fundamental thermodynamic responses of mass flux and gross moist stability as argued in the current study).

Thanks a lot for sharing the thoughtful insights with us. As shown in Figures 1 and 3 in the Response to Reviewer 1, the patterns of high-cloud response to tropical-mean surface temperature change on the centennial time scale resemble those during El Niño and the patterns are closely tied to the surface temperature warming patterns. Long et al. (2016) performed inter-model singular value decomposition (SVD) analysis for 26 climate model simulations under the RCP4.5 scenario and identified several dominant modes of SST warming patterns across the models. We have worked with the leading author Shang-Min Long and her advisor Prof. Shang-Ping Xie to examine the linkage between the leading SST warming patterns and the tropical-wide iris-effect and global-mean precipitation changes. We have found that the global-mean precipitation change, tropical-mean high-cloud fraction change and associated OLR change are highly correlated with the magnitudes of the equatorial-peak SST warming pattern (Figure 4), which is related to the tightening of Hadley Ascent ($\Delta\omega$ in Figure 4a). The models with stronger equatorial peak SST warming tend to have greater reduction of tropical high-cloud fraction and greater global-mean precipitation increase. The zonally asymmetric equatorial SST warming under the RCP4.5 scenario shows an east-west dipole pattern, similar to the "El-Nino" SST anomaly, which is closely associated with the Walker Circulation change. Long et al. (2016) and our new analysis confirm that the large-scale

circulation and high-cloud changes are intimately connected to the projected SST warming patterns. We will report the results in detail in a separate manuscript on the SST-circulation-cloud-precipitation interactions pertaining to the tightening of Hadley ascent. For this manuscript, we have added the discussions on the interactions between SST warming pattern and atmospheric circulation change.

Figure 4. (a) The latitudinal structure of the equatorial peak mode for the changes of SST (in red) and vertical velocity at 500 hPa (in blue). This is the second SVD mode for zonal-mean SST and ω_{500} changes under the RCP4.5 scenario across 26 climate model simulations. (b) Regression of zonal-mean cloud fraction profiles onto the equatorial peak SST warming mode across the 26 models. Red crosses mark the areas where the regression is statistically significant at the 95% level. (c) The relationship between the magnitudes of the equatorial peak SST warming mode and tropical-mean cloudy-sky OLR changes for all models. (d) The relationship between the magnitudes of the equatorial peak SST warming mode and global-mean precipitation changes for all models. All quantities are normalized by global-mean surface temperature changes. **The models that produce stronger equatorial peak SST warming tend to have greater reduction of tropical high-clouds, more OLR and stronger increase of global-mean precipitation.**

2) How much of the zonal mean response is a "tightening" of the Hadley Cell and how much is a reorganisation of the ITCZ into a more zonal configuration?

This is an interesting question. Our definition of the tightening of tropical ascending areas, the dF_{ω}/dT_s , includes the reduction of ascent areas in both zonal and meridional directions. When we decompose it into the zonal-mean and zonally asymmetric

components, we found that the contraction in the zonal-mean component, rather than the zonally asymmetric component, dominates. Therefore, the narrowing of the Hadley Cell plays a greater role in the tightening than the changes of the Walker Circulation, i.e., the re-organization of the ITCZ into a more zonal configuration. It is not clear to us how to better quantify the re-organization of the ITCZ into a more zonal configuration. The study by Wodzicki and Rapp (2016) showed that the narrowing of the ITCZ width occurs in both Central Pacific and Eastern Pacific over the past 36 years (1979-2014), while a stronger narrowing trend is found over the Central Pacific. It would be useful to examine the detailed spatial structure of the tightening in each model following the observational analysis of Wodzicki and Rapp (2016). However, we feel this manuscript focuses on the linkage between the circulation change and cloud/precipitation changes for inter-model spreads. We would defer the analysis of the exact spatial structures of the tightening to a follow-on study.

3) Is there any link to model response in "tightening" and the present day biases in ITCZ location and associated biases in cross equatorial energy transports (e.g. Loeb et al. 2016 Clim. Dyn. doi:10.1007/s00382-015-2766-z)?

This is another interesting point. The linkage between the ITCZ position and cross-equatorial energy transport has been well established in a number of studies, including Loeb et al. (2016). It is natural to connect the model biases in representing the tightening to the biases in climatological ITCZ locations. We have examined the correlation between the interannual dF_{ω}/dT_s and the model biases in the climatological ITCZ positions for the AMIP5 models. The latter were taken from Stanfield et al. (2015) in which the AMIP5 model simulated annual-mean ITCZ centroid positions were evaluated against that of GPCP data. Figure 5 shows the correlation to be 0.42 for the 21 available models. There appears to be a tendency for the models with less northward bias in annual-mean ITCZ location to have a stronger tightening (more negative dF_{ω}/dT_s). We note that the ITCZ centroid position definition in Stanfield et al. (2015) is different from that in Wodzicki and Rapp (2016). Further work is needed to fully explore the dependency of the tightening bias on the climatological ITCZ locations and therefore cross equatorial energy transports in the models. We appreciate the reviewer's thoughtfulness and inspirations. We will continue the investigations in the future.

Figure 5. The relationship between the interannual tightening index, dF_{ω}/dT_s , and the model simulated annual-mean ITCZ centroid position bias relative to the GPCP data for 21 AMIP5 models.

4) If some of the South Pacific Convergence Zone (SPCZ) strays outside the 20S-20N zone this could also influence the amount of the ascending branch of the Hadley circulation that is sampled. Are similar results found for 30S-30N?

Figure 6. The relationship between the centennial tightening index, dF_{ω}/dT_s , and the tropical-mean high-cloud fraction change for the (left) 20°S-20°N and (right) 30°S-30°N averages.

We have conducted the analyses for both tropical (20°S-20°N) and tropical and subtropical (30°S-30°N) averages. All results presented in the paper are valid for both domains while the exact correlation coefficients vary slightly. We choose to show only the 20°S-20°N domain because we focus on the high-clouds and the ascending regions of the Hadley Cell to avoid the influence of mid-latitude storms on the tropical high-cloud fraction (e.g., Hartmann and Michelsen 2002). Figure 6 shows an example of the similarity between the analyses for the two tropical domain averages.

5) Can tightening be diagnosed by considering the probability distribution of vertical motion or by considering changes as a function of surface temperature relative to tropical mean (ie in regime rather than geographical space)?

Thanks for the suggestion. The reduction of tropical ascents corresponds to a decrease of the cumulative frequency of upward motion and an increase in the pdf of neutral vertical motions. The relative warmer surface areas with respect to the tropical-mean surface temperature would decrease, analogous to a conventional El Niño condition when the east-west SST gradient is reduced. Figure 7 shows a typical model response that fits into this expectation. However, we found that the models produce rather complicated shifts in the pdfs and it is difficult to quantify the tightening using the shifts in the pdfs of vertical velocity bins and/or the relative warmth bins. We acknowledge the merits to examine the circulation changes by the regimes but leave the actual investigations to future studies.

Figure 7. The centennial changes in the probability density function (pdf) of (a) vertical pressure velocity at 250 hPa and (b) the surface temperature relative to the tropical-mean for the tropical 20°S-20°N domain.

6) Satellite infrared retrievals sample clear-sky outgoing longwave for cloud-free regions which are drier than cloudy regions and may have stronger sensitivity to temperature changes compared with the models. This of course also affects cloud radiative effect although the influence may be small. Did the authors check this using model data to mimic satellite sampling or do past studies show this effect is small?

We have examined the model biases due to the satellite sampling for many variables, including radiative fluxes, water vapor and cloud properties. Jiang et al. (2012) reported that the satellite sampling biases would cause the model biases around 5%-10%, depending on the different variables. In this study, modeled simulated all-sky OLR and cloud fraction are evaluated against satellite observations. For both quantities, the model biases due to CERES and MODIS satellite sampling frequencies are rather small, well below 5%.

7) Figure 1a/b is rather complicated! I think at the very least a legend would be helpful. It may also be useful to note in the text that negative omega signifies ascent.

We have added legends in Figure 1a/b, kept only the multi-model-means and removed the curves for individual models. We have also added that negative ω signifies ascent.

8) The work of Wodzicki and Rapp (2016) JGR doi:10.1002/2015JD024458 appears to be relevant

Thanks very much for pointing out this paper to us. It is a very relevant study. The observations from GPCP and TRMM provide clear observational evidence of the tightening of tropical ascending regions. We have cited this paper in the main text.

9) *Although differences in surface net longwave do not seem to explain the model diversity, it may be useful to emphasize that longwave radiative cooling is strongly enhanced through reduced net upward longwave flux at the surface as the tropics warm.*

We have added in the text “longwave radiative cooling is strongly enhanced through reduced net upward surface longwave radiation as the surface warms.”

Minor Comments/Corrections

L20 - Abstract: "under warmer"  "in a warmer"

Done.

L41 - LvP is an approximation since snowfall involves the latent heat of fusion also (e.g. Loeb et al. 2016 Clim. Dyn.)

Text modified.

L47/48 - this is slightly vague: global mean precipitation and its variability are determined to a large extent by longwave radiative cooling (LWC) but maybe "primarily constrained" is not quite the best phrase?

The phrase "primarily constrained" is changed to “primarily determined”.

L53 - "in controlling"

Done.

L57 - it should probably be stated that global mean is being discussed?

Added “global-mean”.

L68 - "a human's eye"

Done.

L140 - presumably increased cloud fraction (CF) is linked to higher upper tropospheric humidity as well as the size of the radiator fin

Yes. The sentence is rephrased.

L143 - this seems inaccurate: in the tropics very little surface emission escapes to space: I suggest "lower tropospheric thermal emission" is more precise

Reworded as suggested.

L157 - given the large spatial reorganisation of circulation systems during ENSO, the short term "tightening" could also involve a more zonally uniform ITCZ as well as a physical tightening.

We have modified the sentence.

L164 - the period of CERES data should be stated, presumably 2000-2015? Would a large El Nino like 1997/8 or 2015/16 influence this correlation. A longer model record or combination of CERES/ERBS data may help to test this (e.g. Allan et al. 2014 GRL doi:10.1002/2014GL060962) although I expect this sensitivity is quite robust.

The CERES data we use are from 3/2000 to 10/2015. We have obtained the ERBE data from 1985-1999 for 20°S-20°N averages (36-day averaged, see Lindzen and Choi, 2009 and 2011). We combined ERBE 1995-1999 de-seasonalized anomalies (relative to the 1995-1999 mean seasonal cycle) and CERES 2000-2005 de-seasonalized anomalies (relative to the 2000-2005 mean seasonal cycle) to regress onto corresponding HadCRU4 T_s anomalies. The rate of OLR change with T_s is $4.03 \pm 0.53 \text{ W m}^{-2} \text{ K}^{-1}$, consistent with our estimate using the post-2000 CERES data. We stay with the original results using the longer CERES EBAF data because the latter is deemed more accurate and stable (Loeb et al., J. Clim, 2014, doi: 10.1175/JCLI-D-13-00656.1). Our conclusion regarding the likely hydrological sensitivity at the 95% confidence level is not sensitive to the time periods used for the OLR sensitivity.

L168 - "that most cloud"

Done.

L172 - "when the surface"

Done.

L177 - strengthened descent may be more accurate

Here, we emphasize the reduced ascent at the edge of convective zones. It is not referring to the strengthened descent.

L179 - "is opposed to"  "is the opposite of"?

We have modified "is opposed to" to "is offset by".

L201 - can the reverse also be argued e.g. increased latent heating leads to greater longwave thermal emission?

Yes, we agree with the reviewer that the dynamics and radiation are strongly coupled. It is hard to claim which comes first. We have changed the sentence to highlight the

coupling instead of the cause-and-effect between the latent heating and longwave cooling.

L223 - some further clarification of the "bias removal procedure" may be appropriate here

We have taken into account the reviewer's concern about the robustness of the "bias removal procedure". In the revision, we have abandoned this procedure and used a more robust approach to infer the hydrological sensitivity. Please see the revised section on "Constraining hydrological sensitivity using observations."

L230 - I find it difficult to distinguish cyan from blue

The original Figure 4 has changed. No cyan and blue colors are used together.

L237 - "effective venue"  "effective step/strategy/method"?

We have changed to "an effective step". Thanks for the suggestion.

L240 - do models with unrealistic shortwave absorption sensitivity to temperature (e.g. GISS models) also display unrealistic longwave sensitivities?

Not really. The GISS_e2r model has both unrealistic longwave and shortwave sensitivities, but the INM_cm4 has relatively good performance in longwave but poor performance in shortwave sensitivity. There is no systematic relation. This might be because that the model simulations of high-clouds (more relevant to the cloud longwave effect) and low clouds (more relevant to the cloud shortwave effect) are handled differently. This may also explain the weak correlations between ECS and the high-cloud sensitivities shown in Supplementary Figure 10. The inter-model spread in ECS is primarily contributed by the model diversity in low cloud feedbacks, not the high-cloud feedbacks.

L249 - "under warmer"  "under a warming"

Done.

L260 - "multi"  "multiple"

Done.

L267-270 - I was slightly confused here: are values taken from other studies here or are they calculated from the models used in the present study for RCP8.5?

Following the suggestions by Reviewer 2, we have used the temperature-mediated precipitation change per unit warming from the "abrupt4XCO2" simulations for the hydrological sensitivity. No RCP8.5 simulations are used.

L275 - "promises"  "promise"

Done.

L279, L309 - "same as"  "consistent with"

Done.

L279 - The 1995-2005 period overlaps with less than 5 years of CERES unless a combined ERBS record is used (although clear-sky would probably have to be approximated from reanalysis).

The combined ERBS-CERES record yields a rate of OLR change with surface temperature at $4.03 \pm 0.53 \text{ W m}^{-2} \text{ K}^{-1}$, consistent with our analysis using the post-2000 record from CERES. We stay with post-2000 CERES EBAF data for its reliability and accuracy.

L289 - a reference to fast and slow precipitation responses should be included here.

Done.

L315, L341 - a precise time period would be useful (also for the other datasets rather than having to check supplementary information).

Done.

Fig. 4 - can the short term $LvdP/dT$ s from observations also be shown or is this too uncertain (see further comments below)

Done. See new Figure 4.

L593 - "a-axis"  "x-axis"?

Done.

Supplementary:

L28 - it could be noted that this is close to that expected from the Clausius Clapeyron equation at cold upper troposphere temperatures (e.g. about 14%/K at 200K). How does the AIRS/MLS estimate compare with other UTH data (e.g. Soden et al. 2005 Science, doi:10.1126/science.1115602). Two decimal places is not necessary (e.g. 9.2-15.6%/K is fine).

Thank you for the suggestion. We have added a sentence about the Clausius-Clapeyron relation. The AIRS/MLS combined water vapor data have an uncertainty about 25% for the UTWVP. We have added a sentence about the data uncertainty.

L37, L51-54, L65, L68, L71 - there are errors in the PDF I downloaded ("Error! Reference Source not found").

References corrected.

L52, L66 - "recipitation"  "precipitation"

Done.

L46+ - when considering long term $dOLR/dT_s$ the influence of greenhouse gas increases is important e.g. $(dOLR/dGHG) \times (dGHG/dT_s)$: does this need to be accounted for?

We agree that the increase of greenhouse gases would contribute to the change of OLR in the models. The direct radiative forcing differences need to be accounted for. In the revised version, this is not an issue now because we focus on the temperature-mediated precipitation change (without direct response to radiative forcing) and a new method is employed to apply the emergent constraints on the hydrological sensitivity. This section has been modified extensively.

L54 - the CMAP trends have previously been found to be unreliable due to calibration on atoll data with real trends so I do not think it is appropriate to even consider this dataset. Using the GPCP and TRMM (1997+) data, although perhaps not independent, surely provides a more robust representation and is probably closer to the $dOLR/dT_s$ relationship I think (it would be interesting to compare with CERES OLR for 2000-2015). I found global dP/dT_s of 2.8%/K and a relationship between GPCP global P and ERA interim total atmospheric cooling quite close to unity (Allan et al. 2014 Surv. Geophys. DOI 10.1007/s10712-012-9213-z) for 1988-2008.

We thank the reviewer very much for pointing out the issues of the CMAP data. We agree with the reviewer that the GPCP precipitation change with surface temperature is a reasonable reference to constrain the model simulations. In our revised manuscript, we have employed the GPCP dP/dT_s from 1995-2005 to constrain the model simulations of the same period. As anticipated by the reviewer, the relationship between the observed precipitation sensitivity from GPCP and OLR sensitivity from CERES are consistent with the model relationships. In our new Figure 4a, including or excluding the observed dP/dT_s and $dOLR/dT_s$ data point in the AMIP5 model simulations does not change the correlation and regression slope significantly, suggesting the linkage between $dOLR/dT_s$ and dP/dT_s is quite robust for both models and observations. Therefore, our new emergent constraint on the hydrological sensitivity is based on the combined estimates from both the $dOLR/dT_s$ -inferred dP/dT_s and the GPCP dP/dT_s . Please see the revised section on the emergent constraint and the supplementary section for details.

L65 - "same as"  "consistent with"

Done.

L69 - "orrelation"  "correlation"

Done.

L75 - *I am not completely convinced by this method. Thinking about model "o" for example in Fig. 4, the LvdP/dTs seems to be "over-corrected" due to its deviation from the fit line. A little clarification may just help here.*

Please see the response above for the revised method for the emergent constraint.

L86 - *is there explanation for the lack of relation between dCF/dTs and ECS? High-clouds have a small net radiative effect but I guess also that there is a diverse mix of responses for different meteorological regimes and interannual variations may not be a good proxy for long-term cloud changes?*

Yes. The weak correlations between dCF/dT_s and ECS may result from many factors, including the effect of cloud optical thickness and cloud top height in driving the net cloud radiative effect; the role of low cloud changes in shortwave cloud feedback, and the iris effect on water vapor feedback, as well as the different meteorological regimes, SST warming patterns between the interannual and centennial time scales. Subtle balances for all these factors could make the dCF/dT_s and ECS relations complicated.

Fig. S6 - panel n should be regular size to help comparison and an x-axis should be added to panel l.

Corrected. This is now Supplementary Figure 11.

References:

Allan, R. P., Variability in clear-sky longwave radiative cooling of the atmosphere. *J. Geophys. Res.*, 111, D22105, doi:10.1029/2006JD007304 (2006).

Bony, S, B. Stevens, D. Coppin, T. Becker, K. A. Reed, A. Voigt, and B. Medeiros, Thermodynamic control of anvil cloud amount. *Proc. Nat. Acad. Sci*, 113, 32, 8927–8932 (2016).

Chiang, J. and A. Sobel, Tropical tropospheric temperature variations caused by ENSO and their influence on the remote tropical climate, *Journal of Climate*, **15**, 2616-2631 (2002).

Chung, E.-S., D. Yeomans, and B. J. Soden, An assessment of climate feedback processes using satellite observations of clear-sky OLR, *Geophys. Res. Lett.*, **37**, L02702, doi:10.1029/2009GL041889 (2010).

- Fasullo, J. T., and K. E. Trenberth, A Less Cloudy Future: The Role of Subtropical Subsidence in Climate Sensitivity, *Science*, 338, 792, DOI: 10.1126/science.1227465 (2012).
- Forster, P. M. F., and J. M. Gregory, The climate sensitivity and its components diagnosed from Earth Radiation Budget Data, *J. Clim.*, 19, 39–52, doi:10.1175/JCLI3611.1(2006).
- Hartmann, D.L., and M.L. Michelsen, No Evidence for Iris, *Bull. Amer. Meteor. Soc.* 83, 249-254 (2002).
- Jiang, J. H., et al. (2012), Evaluation of cloud and water vapor simulations in CMIP5 climate models using NASA “A-Train” satellite observations, *J. Geophys. Res.*, 117, D14105, doi:10.1029/2011JD017237 (2012).
- Klein and Hall, Emergent constraints for cloud feedbacks. *Curr. Clim. Change Rep.* 1, 276-287 (2015).
- Lau W. K.-M. & K.-M. Kim, Robust Hadley Circulation changes and increasing global dryness due to CO2 warming from CMIP5 model projections, *Proc. Nat. Acad. Sci.*, 112 (12), 3630–3635, doi: 10.1073/pnas.1418682112 (2015).
- Lindzen, R. S., and Y.-S. Choi, On the determination of climate feedbacks from ERBE data, *Geophys. Res. Lett.*, 36, L16705, doi:10.1029/2009GL039628 (2009).
- Lindzen, R. S., Yong-Sang Choi, On the observational determination of climate sensitivity and its implications, *Asia-Pacific J. Atmos. Sci.*, 47, 4, 377 (2011).
- Long, S.-M., S.-P. Xie, and W. Liu, Uncertainty in tropical rainfall projections: Atmospheric circulation effect and the ocean coupling. *J. Climate*, doi: 10.1175/JCLI-D-15-0601.1 (2016).
- Mauritsen, T. & Stevens, B. Missing iris effect as a possible cause of muted hydrological change and high climate sensitivity in models. *Nature Geosci.* 8, 346–351 (2015).
- Murphy, D. M., S. Solomon, R. W. Portmann, K. H. Rosenlof, P. M. Forster, and T. Wong (2009), An observationally based energy balance for the Earth since 1950, *J. Geophys. Res.*, 114, D17107, doi:10.1029/2009JD012105 (2009).
- Neelin, J. D., D. S. Battisti, A. C. Hirst, F.-F. Jin, Y. Wakata, T. Yamagata, S. Zebiak, ENSO Theory, *J. Geophys. Res.* **103**(C7), 14261-14290 (1998).
- Neelin, J. D. and H. Su, Moist teleconnection mechanisms for the tropical South American and Atlantic sector. *J. Climate*, 18, 3928-3950, doi:10.1175/JCLI3517.1 (2005).

- Ramanathan, V. & Collins, W., Thermodynamic regulation of ocean warming by cirrus clouds deduced from observations of the 1987 El Niño, *Nature* **351**, 27–32 (1991).
- Sherwood, S. C., S. Bony, and J.-L. Dufresne, Spread in model climate sensitivity traced to atmospheric convective mixing, *Nature*, 505, 37–42, doi:10.1038/nature12829 (2014).
- Stanfield R., J. H. Jiang, X. Dong, B. Xi, H. Su, L. Donner, L. Rotstayn, T. Wu, J. Cole and E. Shindo, A Quantitative Assessment of Precipitation Associated With the ITCZ in the CMIP5 GCM Simulations, *Climate Dynamics*, 1-81, doi:10.1007/s00382-015-2937-y, (2015).
- Su, H., J. D. Neelin, and C. Chou, Tropical teleconnection and local response to SST anomalies during the 1997-1998 El Niño, *J. Geophys. Res.*, **106**, No. D17, 20,025-20,043 (2001).
- Su, H., J. D. Neelin, and J. E. Meyerson, Sensitivity of tropical tropospheric temperature to sea surface temperature forcing, *J. Climate*, **16**, 1283-1301 (2003).
- Su, H., and J.H. Jiang, Tropical Clouds and Circulation Changes During the 2006-07 and 2009-10 El Niños, *J. Climate*, 26, 399–413, doi:10.1175/JCLI-D-12-00152.1 (2013).
- Su, H., J.H. Jiang, C. Zhai, T.J. Shen, J.D. Neelin, G.L. Stephens, and L.Y. Yung, Weakening and Strengthening Structures in the Hadley Circulation Change under Global Warming and Implications for Cloud Response and Climate Sensitivity, *J. Geophys. Res.*, 119, 10, 5787–5805, doi:10.1002/2014JD021642, (2014).
- Trenberth, E. T., J. T. Fasullo, C. O. Dell, and T. Wong, Relationships between tropical sea surface temperature and top-of-atmosphere radiation, *Geophys. Res. Lett.*, 37, L03702, doi:10.1029/2009GL042314 (2010).
- Zelinka, M. D., and D. L. Hartmann, The observed sensitivity of high-clouds to mean surface temperature anomalies in the tropics, *J. Geophys. Res.*, 116, D23103, doi:10.1029/2011JD016459, (2011).
- Wodzicki, K. R., and A. D. Rapp, Long-term characterization of the Pacific ITCZ using TRMM, GPCP, and ERA-Interim, *J. Geophys. Res. Atmos.*, 121, 3153–3170, doi:10.1002/2015JD024458 (2016).

Reviewers' comments:

Reviewer #1 (Remarks to the Author):

Review of Revision of Tightening of Hadley ascent key to radiative control 1 on precipitation
2 change in a warmer climate,
by Su, et al.

The authors have shown in their response to review that the interannual variability on which their observational correlations are based is ENSO. When an El Niño occurs, the tropical mean SST is raised, the ITCZ strengthens in the Eastern Equatorial Pacific and the precipitation is more strongly concentrated along the ITCZ, implying a strengthening of the Hadley Cell relative to La Niña conditions. The majority of CMIP5 models produce more warming in the Eastern Equatorial Pacific than elsewhere, and the precipitation increases there a great deal (see e.g. IPCC AR5 Chapter 12). It has been noted by others that the increase in precipitation in the Eastern Pacific is super-Clausius-Clapeyron locally, because the circulation changes give local precipitation increases that are much more than the saturation vapor increase of the tropics as a whole. It would seem then that the emergent constraint is based on the idea that El Niño is a good analog for global warming. There are many reasons to question whether this is a good analogy. Models and observations show that the Hadley Cell contracts during El Niño events, whereas in global warming simulations the Hadley Cell expands a little, despite the tendency toward increased precipitation in the Eastern Pacific in climate models. During an El Niño the Pacific Ocean and the tropics in general are out of equilibrium and relax back toward the alternate state over the course of about a year. Transient climate change is also a sequence of un-equilibrated states, and maybe the enhanced warming in the east Pacific in response to greenhouse gas increases is a robust scientific conclusion, but IPCC AR5 concluded that this response is not sufficiently robust to be given a likely rating, in part because the control simulations of ENSO in most CMIP5 models are still pretty bad.

The argument is that the spread of the cloud response to ENSO is relevant to constraining the spread in global warming simulations, which could conceivably be useful, even if the models are wrong about the El Niño-like response to global warming.

To what extent do that authors believe that their conclusions are dependent on the El Niño global warming response, and to what extent do they believe that basic physics would provide a reduction of high cloud area in response to global warming, irrespective of its spatial structure? Thermodynamic arguments suggest a reduction in convective mass flux irrespective of the spatial structure of the convection patterns. Can they do the job, or is an El Niño response needed?

The authors should note that the mechanisms they are talking about are already incorporated into current climate models, and so there should be no inference that climate is less sensitive than current models indicate. The new parts suggesting that models underestimate the cloud area response to tropical mean SST must be tempered by the realization that the models might just not simulate a very good ENSO cycle, which might have nothing to do with the fidelity of the cloud physics, although I think it is fair to say that the methods used to predict the amount of ice in tropical clouds are primitive and uncertain.

The reviewer thanks the authors for clarifying the difference between tropical estimates of $dOLR/dTs$ and global estimates, which are closer to $2.0 \text{ Wm}^{-2}/\text{K}$, an interesting difference. This is probably mostly based on the ENSO response?

Specific comments:

Lines 27-30: What does this sentence mean? 95% probability of what? Does this mean that it is 95% certain that the change will be larger than the mean precipitation, or 95% certain that the actual change in the precipitation will be greater than the mean of CMIP5? And where are you

talking about? The tropical mean or the maximum over the East Pacific?

39-40 I think 1% to 3% is a better estimate for the true uncertainty. Removing the sensitivity due to forcing uncertainty is a nicety that may be useful for theoretical reasons, but not in practice.

250: Equally likely the shortwave effects of the high clouds themselves cancel the longwave effects of the high clouds changes.

272: Is this because the interannual variations of tropical SST have the structure of ENSO, or are some more basic physical processes at work? Is the El Niño response of the CMIP5 models necessary to produce this relatively large tropical sensitivity of OLR to surface temperature?

285-322 The smaller OLR to T_s connection in models could be related to their poor job of simulating ENSO than to their representation of the physics that control global warming. I think the proposed emergent constraint is a rather loose-jointed one, with many uncertain and interacting parts.

Reviewer #3 (Remarks to the Author):

Second review of "Tightening of Hadley ascent key to radiative control on precipitation change in a warmer climate" by Su et al.

The authors have addressed all my questions and suggestions and I consider that this work is a potentially important contribution to the understanding of hydrological response to climate change. The only substantial outstanding question I have relates to the scaling of cloud fraction (CF) and ascent area ($F\omega$) to surface temperature (T_s) for individual models between interannual and centennial scales.

In Figures 2 and 3 there is good apparent correspondence between dCF/dT_s and $dF\omega/dT_s$ and between $dOLR/dT_s$ and dCF/dT_s when considering the range of model responses. Although the relationships are consistent for both interannual and climate change time-scales, consistency between timescales does not seem to be the case for individual models. For example, model g has a negative dCF/dT_s and $dF\omega/dT_s$ scaling for centennial time-scales but not for interannual. This seems to suggest that the present day sensitivity is not relevant for the future response, at least when considering model by model.

The centennial-scale changes also include fast responses to radiative forcing which may explain the contrasting scalings but if the mechanisms proposed are operating consistent temperature mediated and interannual cloud fraction and dynamical responses should be demonstrated and further linked to the centennial responses which are more closely aligned with the future projected changes in precipitation patterns and therefore impacts.

These points may just need a little further clarification with supporting evidence. Below are a number of additional minor suggestions which I consider should also be addressed.

Abstract: "the 95% likely predictions" needs rephrasing (at the 95% confidence level?)

L33 ecosystem  ecosystems

L35 "that requires accurate simulations by climate models." is vague and can be removed

L38 "temperature-mediated global-mean precipitation" refers to temperature-only response; this could be made clear here but do the past studies referred to also separate out this temperature-

only related response?

L46 the latent heat of fusion is not the same as vaporisation

L55 "despite that"  "although"

L61 the long-term (centennial) coupled relationships will also strongly depend on fast responses to radiative forcings and so will depend upon the precise radiative forcing mix and time series

L111 "cartoon"  "schematic"

L117 The recent study of He and Soden (2016) Nature Climate Change, consistent with past work by Bony et al. (2013) Nature Geosci., indicates that there is a strong fast dynamical response to CO2 radiative forcing leading to drying of subtropical zones. The authors may wish to consider if this is relevant.

L134 "strong resemblance between the relationships on the two time scales provides a physical basis for a measure of 'emergent constraint'" - I think "physical basis" is too strong since these merely show similarity and could be determined by different processes (such as dynamical tendency toward more El Nino like state dependent on regional SST feedbacks). I suggest toning this statement down or adding a caveat

L210 "rate rates"  "rain rates"?

L252 "On Figure 3a"  "In Figure 3"?

L304 HadCRUT4

L311 The interannual and long-term T-mediated dP/dTs responses are correlated but what about the dOIR/dTs and dF_ω/dTs which are key to the proposed mechanisms?

L338 "ascent"

L343 "by about 5 times." The spread is reduced from 1.1 %/K to 0.3 %/K, one quarter of the overall range in terms of percentage (I assume 5 times refers to Wm⁻²K⁻¹ but quoting the range as a percentage of the initial range may be clearer).

L345 "all on the upper half of the CMIP5 model ensemble." - can it be said here that the cloud feedback on hydrological sensitivity (e.g. O'Gorman et al. 2012) causes these models to produce a stronger response?

L249 it could be noted that this is a tropic-wide response but that regionally, fast response to radiative forcing may be important (e.g. He and Soden 2016).

L376 it should be explained how the Temperature-mediated response is calculated (e.g. by regression over time over a certain year range of the simulation) and how this can be related to centennial regional response, more relevant for impacts

L395 "same as"  "consistent with"

L398 "Hadley Centre and Climate Research Unit"

L403 "version 2.8"

Supplementary L25 it would be useful to redefine LWC again here (and other variables e.g. SWA)

Supplementary L94 HadCRUT4

Response to Reviewers

Comments from Reviewers are marked in red and italicized. Our responses are in blue. The page numbers and line numbers in the response below refer to those in the track-changed manuscript.

Summary:

We thank the reviewers' thoughtful comments and constructive suggestions. We have addressed the reviewers' comments point-by-point with additional analyses and discussions in the text. The major additions include the analysis of temperature-mediated sensitivities for circulation, cloud fraction and OLR from about 20 models' abrupt4×CO₂ experiments. The results are consistent with the interannual and centennial sensitivities. We have also examined the relationship between interannual and centennial dP/dT_s under RCP4.5 and found a statistically significant positive correlation, suggesting that the models that underestimate the interannual dP/dT_s tend to have weaker centennial dP/dT_s , and vice versa, although the fast response to direct CO₂ forcing adds noticeable deviations to the positive relation between interannual and centennial dP/dT_s . The new analyses results are consistent with the original conclusions. We have added the discussions of the new results in the revised manuscript, with new figures in Supplementary Information.

Response to Reviewer 1

We greatly thank the reviewer for sharing with us his/her profound understanding of the tropical dynamics and thermodynamics. His/her thoughtful comments and deep insights have helped us interpret the results more accurately. In particular, the reviewer raised questions about the analogy between ENSO and global warming, the realism of the common El Niño-like SST warming pattern in the climate models and the dependency of cloud and precipitation responses on the spatial structure of the warming pattern. He/she also asked whether the source of model biases is the uncertainty in ice cloud physics or imperfect simulations of ENSO cycle. Our detailed responses are inserted after the reviewer's comments. The summary response is itemized below.

1. We agree with the reviewer that ENSO is not a good analogy to global warming in many aspects. Our study uses El Niño warming as a test case to establish the physical pathways that link circulation, cloud fraction, longwave radiation, and precipitation change. The sentence on Page 4, Line 97-99 is rephrased.
2. We think that the tightening of the equatorial ascending regions and the decrease of tropical-mean high cloud fraction in a warmer climate are governed by fundamental thermodynamics and tropical dynamics. The increase of static stability associated with surface warming (Bony et al., 2016) and the upped-ante

mechanism (Neelin et al., 2003) are possible mechanisms for the tightening of tropical ascent and the high cloud reduction. These mechanisms do not require the El Niño-like SST warming patterns. Bony et al. (2016) showed that a tropical-wide “iris effect” could occur with prescribed uniform SST warming. When ocean-atmosphere is coupled, the circulation and cloud changes are intimately tied to spatial structure of SST. We have elaborated this on Page 7, Line 184-191, and Page 9, Line 267-270.

3. Although it is debatable whether the El Niño-like warming pattern in response to greenhouse gas increase is realistic, the analogous relationships between the circulation, cloud, radiation, and precipitation on the interannual and centennial time scales suggest that similar physical processes are at work for the two phenomena. Hence, constraining interannual sensitivities is useful to constrain long-term sensitivities. This is discussed on Page 4, Line 97-99; Page 8, 221-225.
4. Since our interannual sensitivities are derived from the AMIP experiments in which observed SSTs are prescribed in all models, the model differences in cloud and precipitation responses are not caused by the models’ inability in simulating the ENSO cycle. Inaccurate cloud parameterizations could be an important source for the inter-model spreads on the interannual sensitivities. On the other hand, in RCP4.5 simulations where coupled atmosphere-ocean interactions are considered, the model deficiencies in simulating the ENSO cycles may contribute to the model differences in the SST spatial distributions and thus circulation and cloud responses on the centennial time scales. We have added the discussions about the possible contributions of ENSO cycle simulations to the inter-model spreads on Page 10, Line 318-319 and Page 11, Line 326-327.

“The authors have shown in their response to review that the interannual variability on which their observational correlations are based is ENSO. When an El Niño occurs, the tropical mean SST is raised, the ITCZ strengthens in the Eastern Equatorial Pacific and the precipitation is more strongly concentrated along the ITCZ, implying a strengthening of the Hadley Cell relative to La Niña conditions. The majority of CMIP5 models produce more warming in the Eastern Equatorial Pacific than elsewhere, and the precipitation increases there a great deal (see e.g. IPCC AR5 Chapter 12). It has been noted by others that the increase in precipitation in the Eastern Pacific is super-Clausius-Clapeyron locally, because the circulation changes give local precipitation increases that are much more than the saturation vapor increase of the tropics as a whole. It would seem then that the emergent constraint is based on the idea that El Niño is a good analog for global warming. There are many reasons to question whether this is a good analogy.

We agree with the reviewer that El Niño and global warming are different in many aspects. Our study uses the interannual warming as a test case to establish the physical pathways from the circulation and cloud changes to global-mean precipitation change. Our analysis results show that the tightening of Hadley ascent and the shrinkage of tropical high cloud fraction occur similarly under global warming and during the interannual variations. Our study focuses on these common features and does not intend to generalize that ENSO and global warming are analogous in every aspect. Our analysis

results show that the interannual and centennial sensitivity rates are positively correlated but with considerable scatter, and their magnitudes are very different. The original text on Page 4, Line 61-63 is modified.

Models and observations show that the Hadley Cell contracts during El Niño events, whereas in global warming simulations the Hadley Cell expands a little, despite the tendency toward increased precipitation in the Eastern Pacific in climate models.

Under global warming, the extra-tropical boundaries of the Hadley Cell expand, but the equatorial ascending regions contract, as shown in Figure 1 in the manuscript and previous studies such as Su et al. (2014) and Lau and Kim (2015). The contraction of equatorial ascents occurs simultaneously with the poleward expansion of descending regions in the subtropics.

During an El Niño the Pacific Ocean and the tropics in general are out of equilibrium and relax back toward the alternate state over the course of about a year. Transient climate change is also a sequence of un-equilibrated states, and maybe the enhanced warming in the east Pacific in response to greenhouse gas increases is a robust scientific conclusion, but IPCC AR5 concluded that this response is not sufficiently robust to be given a likely rating, in part because the control simulations of ENSO in most CMIP5 models are still pretty bad. The argument is that the spread of the cloud response to ENSO is relevant to constraining the spread in global warming simulations, which could conceivably be useful, even if the models are wrong about the El Niño-like response to global warming.

We agree with the reviewer that it is debatable whether the El Niño-like warming pattern is realistic in the climate model projections, because there are a lot of uncertainties in the models that could affect the SST distributions and evolutions, such as inaccurate representation of clouds, winds, surface fluxes and ocean dynamics. However, as the reviewer stated, it is still useful to constrain the model spread in global warming simulations based on the fidelity of present-day model simulations on the interannual time scale because of the similar physical pathways from circulation and cloud changes to global-mean precipitation change on the two time scales. How accurate is the El Niño-like warming pattern merits further investigation but is beyond the scope of this study.

To what extent do that authors believe that their conclusions are dependent on the El Niño global warming response, and to what extent do they believe that basic physics would provide a reduction of high cloud area in response to global warming, irrespective of its spatial structure? Thermodynamic arguments suggest a reduction in convective mass flux irrespective of the spatial structure of the convection patterns. Can they do the job, or is an El Niño response needed?

As mentioned earlier in the summary response, Bony et al. (2016, PNAS) analyzed three climate model simulations under idealized settings: aqua-planet, no Earth's rotation, uniform solar insolation and globally uniform SST. They varied the constant SSTs from 295 K to 305 K and found that the anvil cloud amount would decrease with surface

warming under radiative-convective equilibrium. The decrease of anvil clouds is driven by the reduction of radiatively driven upper tropospheric mass convergence. The latter is caused by the increase of static stability with surface warming. Their study suggests that the tropical-wide “iris effect” would occur even when there is no spatial gradient of SST. They also showed that the iris effect happens when cloud radiative effect is turned off, but the magnitude of the iris effect is amplified when cloud radiative effect is turned on.

In our study, we examine both the AMIP simulations and the coupled simulations (the RCP4.5-historical runs and the abrupt4×CO₂ runs). In AMIP simulations, the prescribed SST distributions during El Niño, including tropical-wide warming and spatial gradient of SST, may act together to induce circulation and cloud changes. The limited moisture supply at the convective margin causes the weakening of convective activities and thus the contraction of ascending areas via the upped-ante mechanism (Neelin et al., GRL, 2003). Thus, the tropical-wide “iris effect” would happen irrespective of spatial structure of SST anomalies. In coupled ocean-atmosphere simulations, the changes of atmospheric circulation and clouds could induce certain spatial structures of SST distributions. Such SST spatial structures could further enhance the narrowing of Hadley ascent and the reduction of high clouds. The direct heating of the atmosphere and land surface resulted from the fast response to CO₂ could also promote subtropical drying (He and Soden, Nature Climate Change, 2016). Therefore, we think the “iris effect” and the tightening of the Hadley ascent could happen by simple thermodynamic argument (e.g., static stability and upped-ante mechanisms) regardless of SST patterns. Our study uses El Niño warming to test the physical processes in response to surface warming. The approximately similar SST warming pattern under global warming and during El Niño further facilitates the linkage but is not a prerequisite. In the revised manuscript, we have clarified the mechanisms responsible for the tightening and “iris effect” on Page 7 Line 184-191 and Page 9, Line 267-270.

The authors should note that the mechanisms they are talking about are already incorporated into current climate models, and so there should be no inference that climate is less sensitive than current models indicate.

The basic mechanisms such as the increasing static stability with surface warming and the “upped-ante” mechanism to create the tightening of Hadley ascent and the decrease of high clouds are included in climate models, but the **magnitude** of these mechanisms and the interactions of these mechanisms with cloud physics and cloud radiative effects are not accurately represented in the models. Our analysis shows that there is a large spread in the magnitudes of the circulation and cloud responses in the CMIP5 models. Some models are closer to the observations and some models deviate from the observations significantly. Hence, our study uses state-of-the-art observations to identify the “better performing” models and to infer the more realistic future projections based on those “better performing” models. Figures 3 and 4 and associated discussions have addressed this issue.

The new parts suggesting that models underestimate the cloud area response to tropical mean SST must be tempered by the realization that the models might just not simulate a

very good ENSO cycle, which might have nothing to do with the fidelity of the cloud physics, although I think it is fair to say that the methods used to predict the amount of ice in tropical clouds are primitive and uncertain.

We would like to point out that the interannual cloud and circulation sensitivities to surface temperature analyzed in this study are based on the AMIP simulations in which observed SSTs are prescribed in all models. Therefore, it is not the representation of ENSO cycle that causes the model differences in the interannual cloud and circulation responses. We have clarified the use of AMIP simulations on Page 4, Line 91-94. We agree with the reviewer that the ice clouds in the tropics are poorly simulated in the models and uncertainties in ice cloud physics could be one of the important sources for the model errors. In RCP4.5-historical simulations, the model simulations of ENSO cycles due to inaccurate representation of ocean-atmosphere interaction can also contribute to the model discrepancies, in addition to the errors in cloud physics. We have added this point in the discussions in the text, Page 10, Line 318-319 and Page 11, Line 326-327.

The reviewer thanks the authors for clarifying the difference between tropical estimates of $dOLR/dT_s$ and global estimates, which are closer to $2.0 \text{ Wm}^{-2}/\text{K}$, an interesting difference. This is probably mostly based on the ENSO response?

The observed $dOLR/dT_s$ is based on the regression slope of tropical-mean monthly OLR anomalies onto the T_s anomalies. Choi et al. (2014) analyzed in detail the difference between the tropical-mean and global-mean $dOLR/dT_s$ with various lags between OLR and T_s . They found that non-feedback noise such as anomalous clouds induced by extratropical weather systems contribute to the tropical-mean and global-mean $dOLR/dT_s$ difference. In addition, the global-mean $dOLR/dT_s$ peaks when OLR lags T_s by 2-4 months, while the tropical-mean $dOLR/dT_s$ maximizes at zero lag (see Figure 1 in Choi et al. 2014 below). Our study examines only the zero-lag regressions. Please also note our estimates of longwave feedback include the Planck response. Choi et al. (2014) also found that the large magnitude of tropical-mean $dOLR/dT_s$ is not affected when strong ENSO events are excluded in the regression calculations.

Figure 1 from Choi et al. (2014). The lagged linear correlation coefficient (a) and regression slope (b) of ΔR versus ΔT_s for shortwave (blue) and longwave (red) radiation; the thick solid line indicates global data, and the thick dashed line indicate 20°S-20°N. The thin red line indicates longwave radiation where Planck response is excluded. In the paper, positive signs are used for upward fluxes.

Specific comments:

Lines 27-30: What does this sentence mean? 95% probability of what? Does this mean that it is 95% certain that the change will be larger than the mean precipitation, or 95% certain that the actual change in the precipitation will be greater than the mean of CMIP5? And where are you talking about? The tropical mean or the maximum over the East Pacific?

Thanks for pointing out the ambiguity of the phrases. The precipitation sensitivity dP/dT_s in the abstract and throughout the manuscript refer to the global-mean precipitation change per unit surface warming. The “95% likely predictions” meant that the predicted dP/dT_s under global warming by the models that agree with the observation-based interannual dP/dT_s at the 95% confidence level, i.e, within 2σ of the observational estimate. We have rephrased the sentence in Line 28-30 to be “We find that the five models that agree with the observation-based interannual dP/dT_s all predict dP/dT_s under global warming higher than the ensemble mean dP/dT_s from the ~20 models analyzed in this study.”

39-40 I think 1% to 3% is a better estimate for the true uncertainty. Removing the sensitivity due to forcing uncertainty is a nicety that may be useful for theoretical reasons, but not in practice.

We agree with the reviewer that the temperature-mediated dP/dT_s is useful for theoretical studies but the full dP/dT_s that includes the forcing uncertainty has practical value. We thus modify the text to mention both full dP/dT_s range and the temperature-mediated dP/dT_s range.

250: Equally likely the shortwave effects of the high clouds themselves cancel the longwave effects of the high clouds changes.

Thanks for pointing this out. We have added in the sentence: “and the shortwave effects of high cloud changes may cancel their longwave effects”.

272: Is this because the interannual variations of tropical SST have the structure of ENSO, or are some more basic physical processes at work? Is the El Niño response of the CMIP5 models necessary to produce this relatively large tropical sensitivity of OLR to surface temperature?

Our estimate of interannual $dOLR/dT_s$ is based on the regression slope of the observed monthly-mean tropical-mean ERBE and CERES OLR data against HadCRUT4 T_s over the past two decades. Out of the total ~4 W/m²/K for the tropical-mean $dOLR/dT_s$, the clear-sky $dOLR/dT_s$ accounts for about 3 W/m²/K and the cloud longwave radiative effect accounts for about 1 W/m²/K. Mauritsen and Stevens (2015) showed that the observed $dOLR/dT_s$ is greater than the combined Planck feedback and water vapor feedback of 2.12 W/m²/K for the tropical 20°S to 20°N, so that it is clear that the decrease of high-level cloud with surface warming play an important role in producing this large

sensitivity of $dOLR/dT_s$. We have discussed this on Page 15, Line 430-432. Based on Bony et al. (2016), even uniform SST warming can produce a decrease of tropical anvil clouds due to the increase of static stability. Therefore, we think that the El Niño SST warming pattern is not a necessary condition to produce the large $dOLR/dT_s$. We recognize that it is difficult to quantify the exact contribution of the SST spatial structure to the large $dOLR/dT_s$ using the observations alone or the archived CMIP5 model simulations. Carefully designed model experiments may help to elucidate it. We defer further quantifications to future studies.

285-322 The smaller OLR to T_s connection in models could be related to their poor job of simulating ENSO than to their representation of the physics that control global warming. I think the proposed emergent constraint is a rather loose-jointed one, with many uncertain and interacting parts.

The interannual sensitivities including $dOLR/dT_s$ analyzed in this study are derived from AMIP simulations in which observed SSTs are prescribed; the low biases in the models' interannual $dOLR/dT_s$ are thus not caused by the model representation of ENSO cycles. For the biases in $dOLR/dT_s$, the model representations of water vapor and cloud feedbacks are most relevant. We have shown that the simulated water vapor is not the primary error source (see Supplementary Information), and the underestimate of the “iris effect” could explain about 50% of the across-model variance in the tropical-mean $dOLR/dT_s$ (Figure 3a). While we agree with the reviewer that the model deficiencies in simulating ENSO cycle could contribute to the model diversity in the coupled simulations, we believe that the poor representation of the cloud and circulation responses to surface warming is a dominant factor that drives the inter-model spread in $dOLR/dT_s$ under global warming. The mechanisms we have identified have unambiguous consequences in affecting atmospheric longwave cooling rate and thus hydrological sensitivity. Our study clearly points out an area for model improvements and it would have direct impact on reducing the uncertainties in model predictions of future precipitation change. In Figure 4 and corresponding discussions on Page 16-17, we show that the low biases in the tropical-mean $dOLR/dT_s$ lead to the low biases in the global-mean dP/dT_s , and the interannual dP/dT_s is highly correlated with the dP/dT_s under global warming. Our emergent constraints are physically based and derived from robust statistical relationships between the changes of tropical circulation, cloud, radiation and global-mean precipitation. The close interactions among these factors are the essence of the global hydrological cycle, a key point emphasized in this study. We have clarified the discussions in this section on Page 16-17 and in Conclusions on Page 18-19.

Response to Reviewer 3

We thank the reviewer's constructive suggestions. We have added the analysis about the temperature-mediated sensitivities derived from the abrupt4×CO₂ experiments. We have also examined the relationship between interannual and centennial dP/dT_s . All the new results are consistent with the original conclusions and further strengthen the manuscript. A point-by-point response is provided below.

The authors have addressed all my questions and suggestions and I consider that this work is a potentially important contribution to the understanding of hydrological response to climate change. The only substantial outstanding question I have relates to the scaling of cloud fraction (CF) and ascent area (F_{ω}) to surface temperature (T_s) for individual models between interannual and centennial scales.

In Figures 2 and 3 there is good apparent correspondence between dCF/dT_s and dF_{ω}/dT_s and between $dOLR/dT_s$ and dCF/dT_s when considering the range of model responses. Although the relationships are consistent for both interannual and climate change time-scales, consistency between timescales does not seem to be the case for individual models. For example, model g has a negative dCF/dT_s and dF_{ω}/dT_s scaling for centennial time-scales but not for interannual. This seems to suggest that the present day sensitivity is not relevant for the future response, at least when considering model by model.

The reviewer is very observant and we appreciate his/her pointing out the caveat. We note that the interannual dP/dT_s and $dOLR/dT_s$ are significantly correlated with the temperature-mediated and centennial dP/dT_s and $dOLR/dT_s$ ($R = 0.5 - 0.6$), respectively, when all models are considered (see Figure 4, Supplementary Figure 13 and Supplementary Figure 15a below). There is noticeable scatter in these correlations, suggesting that other factors not analyzed in the study are at play. For dCF/dT_s and dF_{ω}/dT_s , the interannual and long-term rates are less correlated ($R \approx 0.2$), as the reviewer sharply pointed out based on the individual models in Figure 2 and 3. We think that the definitions of high cloud fraction and tropical circulation tightening indices may be more sensitive to the mean climate states and temporal scales of variability than the energy flux terms such as OLR and precipitation. For example, the high cloud fraction definition according to the ISCCP standard in this study may not be the best choice to link with OLR change or the radiatively driven upper level mass convergence (i.e., convective detrainment). Tropical circulation index under different climate states might need to use ω at different heights or different threshold value for upward motions tied to convective detrainment ($\omega < 0$ is used in the paper). On the other hand, the energy flux variables, such as dP/dT_s and $dOLR/dT_s$, are more robustly defined so that their variabilities in the models are more consistently represented between short-term and long-term time scales. Thus they are better suited as emergent constraints for future climate change predictions than dCF/dT_s and dF_{ω}/dT_s . Figure 2 and Figure 3 reveal that the dominant processes that control the inter-model spreads in $dOLR/dT_s$ are dF_{ω}/dT_s and dCF/dT_s for both interannual variability and global warming, while Figure 4 and Supplementary Figure 1 disclose that $dOLR/dT_s$ contributes predominantly to the inter-model spread in dP/dT_s on both time scales. The circulation and cloud fraction analyses (Figure 2 and Figure 3) establish the physical pathways but are not used as emergent constraints on hydrological sensitivity.

As the reviewer pointed out, the fast response to direct CO_2 forcing, which is independent of surface temperature change, may also affect the relative scaling of centennial sensitivities compared to interannual sensitivities. We find that the inter-model spread in temperature-mediated sensitivities accounts for a large fraction of the inter-model spread in centennial sensitivities (as shown in Supplementary Figure 15b below), although the

fast response to direct CO₂ forcing can create further diversity in the models' centennial sensitivities.

Supplementary Figure 15. Relationships between interannual, temperature-mediated and centennial $dOLR/dT_s$. (a) The temperature-mediated $dOLR/dT_s$ scattered against the interannual $dOLR/dT_s$. (b) The centennial $dOLR/dT_s$ scattered against the temperature-mediated $dOLR/dT_s$. Multi-model-means are marked in solid circles. The least-squares linear regression lines are drawn.

The centennial-scale changes also include fast responses to radiative forcing which may explain the contrasting scalings but if the mechanisms proposed are operating consistent temperature mediated and interannual cloud fraction and dynamical responses should be demonstrated and further linked to the centennial responses which are more closely aligned with the future projected changes in precipitation patterns and therefore impacts.

Yes, we agree with the reviewer that the centennial changes include the fast response to radiative forcing and the slow response to surface warming, and the temperature mediated circulation and cloud responses should be examined. Following his/her suggestion, we have added the analysis of 20 models' abrupt4×CO₂ experiments. The temperature-mediated responses are derived from the regression slopes of annual-mean tropical or global mean quantities against annual-mean T_s change relative to the piControl run for the first 150 years of abrupt4×CO₂ experiments. Counterparts of Figure 2 and 3 for temperature-mediated sensitivities are shown in Supplementary Figure 6 (shown below). The new analysis results further support our original conclusions: the tightening of Hadley ascent is highly correlated with the decrease of tropical-mean high cloud fraction ($R = 0.62$) and the latter is the primary driver of the inter-model spread in temperature-mediated $dOLR/dT_s$ ($R = -0.71$). We note that the temperature-mediated sensitivities are correlated with the centennial sensitivities although they are clearly different in magnitude (see preceding Supplementary Figure 15), and the relative scaling does not hold for individual models' interannual and temperature-mediated sensitivities (see the response earlier).

Supplementary Figure 6. Relationships between the inter-model spreads in the tightening of Hadley ascent, the tropical high cloud fraction sensitivity, and the OLR sensitivity for the temperature-mediated rates. (a) The tropical-mean dCF/dT_s scattered against the change of the tropical ascending areas per unit surface warming based on the upward velocity at 250 hPa, dF_w/dT_s . (b) The tropical-mean $dOLR/dT_s$ scattered against the tropical-mean dCF/dT_s . The sensitivities are derived from the abrupt4xCO₂ experiments using the linear regression method. Multi-model-means are marked in solid colored circles. The least-squares linear regression lines are drawn.

At the reviewer's request, we have also applied the interannual observation-based dP/dT_s to constrain the model projections of centennial precipitation changes because the centennial dP/dT_s is directly relevant to societal impacts. We find that the interannual dP/dT_s is positively correlated with the centennial dP/dT_s with $R = 0.55$ for the 20 models examined, suggesting that the temperature-mediated dP/dT_s dominates the inter-model spread in the centennial dP/dT_s . The models with larger interannual dP/dT_s tend to have larger centennial dP/dT_s , although not every individual model obeys this scaling, partly because of the fast response to direct CO₂ forcing.

These points may just need a little further clarification with supporting evidence. Below are a number of additional minor suggestions which I consider should also be addressed.

We thank again the reviewer's constructive suggestions. These new analyses further strengthen our original conclusions. The caveat about the relative scaling of individual models is also noted in the revised manuscript on Page 17, Line 497-501.

Specific Comments/Corrections

Abstract: "the 95% likely predictions" needs rephrasing (at the 95% confidence level?)

Yes, we have rephrased the sentence to be "We find that the five models that agree with the observation-based interannual dP/dT_s all predict dP/dT_s under global warming higher than the ensemble mean dP/dT_s from the ~20 models analyzed in this study."

L33 ecosystem  ecosystems

Done.

L35 "that requires accurate simulations by climate models." is vague and can be removed

Rephrased to be “that represents the sensitivity of the climate system to global warming”.

L38 "temperature-mediated global-mean precipitation" refers to temperature-only response; this could be made clear here but do the past studies referred to also separate out this temperature-only related response?

Rephrased.

L46 the latent heat of fusion is not the same as vaporization

You are right. We have added “roughly” in the sentence and avoid using the latent heat of fusion for simplicity.

L55 "despite that"  "although"

Done.

L61 the long-term (centennial) coupled relationships will also strongly depend on fast responses to radiative forcings and so will depend upon the precise radiative forcing mix and time series

Thanks for pointing out this. Since we have analyzed the temperature-mediated response in addition to the centennial relationships, we use global warming relations to refer to the common behaviors for both centennial and temperature-mediated responses. The original Line 61-64 is modified.

L111 "cartoon"  "schematic"

Done.

L117 The recent study of He and Soden (2016) Nature Climate Change, consistent with past work by Bony et al. (2013) Nature Geosci., indicates that there is a strong fast dynamical response to CO2 radiative forcing leading to drying of subtropical zones. The authors may wish to consider if this is relevant.

Thanks for pointing out the relevant studies. We have added the discussion of this mechanism and the reference on Page 8, Line 213-214.

L134 "strong resemblance between the relationships on the two time scales provides a physical basis for a measure of 'emergent constraint'" - I think "physical basis" is too strong since these merely show similarity and could be determined by different processes

(such as dynamical tendency toward more El Nino like state dependent on regional SST feedbacks). I suggest toning this statement down or adding a caveat

We have toned down the statement.

L210 "rate rates"  "rain rates"?

Done.

L252 "On Figure 3a"  "In Figure 3"?

Done.

L304 HadCRUT4

Done. Thanks!

L311 The interannual and long-term T-mediated dP/dT_s responses are correlated but what about the $dOIR/dT_s$ and dF_{ω}/dT_s which are key to the proposed mechanisms?

As shown earlier, the interannual and long-term $dOLR/dT_s$ are correlated, but the correlations for interannual and long-term dF_{ω}/dT_s are not very good, about ~ 0.2 . This is understandable as there are larger scatters in Figure 2 than in Figure 3. We speculate that the circulation tightening index may be sensitive to exact vertical height (we use 250 hPa in the paper) and the threshold value for ω ($\omega < 0$ is used in the paper) when defining the tropical ascending area that corresponds to high cloud amount. In a warmer climate, the vertical height for the index may need to be adjusted. The correlation coefficients between interannual and long-term dF_{ω}/dT_s would vary when different height or threshold value for ω is used. Since we have demonstrated the linkage between the tightening of Hadley ascent and the decrease of high cloud fraction on interannual and centennial time scales separately, and the short-term and long-term sensitivities for OLR and precipitation are significantly correlated, we think the coupled circulation-cloud-radiation-precipitation relations are applicable to both time scales. We use dF_{ω}/dT_s and dCF/dT_s correlations (Figure 2) to explain the physical mechanisms for the "iris effect", but do not use them as the emergent constraints to infer future precipitation changes, partly because of the inherent uncertainties in their definitions and the lack of direct observations of ω as well as the incompatible cloud fraction measurements from different satellite sensors. In comparison, OLR and precipitation are more robustly observed and consistently represented in the models across the time scales. In the revised manuscript, we use the combined interannual dP/dT_s and $dOLR/dT_s$ observations to constrain the models' hydrological sensitivity. We have clarified this on Page 17, Line 497-501.

L338 "ascent"

Done.

L343 "by about 5 times." The spread is reduced from 1.1 %/K to 0.3 %/K, one quarter of the overall range in terms of percentage (I assume 5 times refers to Wm-2K-1 but quoting the range as a percentage of the initial range may be clearer).

We have modified “by about 5 times” to “from the original range of 1.1%/K to 0.3%/K”.

L345 "all on the upper half of the CMIP5 model ensemble." - can it be said here that the cloud feedback on hydrological sensitivity (e.g. O'Gorman et al. 2012) causes these models to produce a stronger response?

Yes, the sentence is added.

L349 it could be noted that this is a tropic-wide response but that regionally, fast response to radiative forcing may be important (e.g. He and Soden 2016).

Here, we emphasize the increase of wet-area precipitation associated with the tightening of Hadley ascent. The subtropical drying due to the fast response to direct CO₂ forcing is discussed on Page 8, Line 213-214. The reference He and Soden (2016) is cited there.

L376 it should be explained how the Temperature-mediated response is calculated (e.g. by regression over time over a certain year range of the simulation) and how this can be related to centennial regional response, more relevant for impacts.

Added.

L395 "same as"  "consistent with"

Done.

L398 "Hadley Centre and Climate Research Unit"

Done.

L403 "version 2.8"

Done.

Supplementary L25 it would be useful to redefine LWC again here (and other variables e.g. SWA)

Done.

Supplementary L94 HadCRUT4

Done.

References:

Bony, S, B. Stevens, D. Coppin, T. Becker, K. A. Reed, A. Voigt, and B. Medeiros, Thermodynamic control of anvil cloud amount. *Proc. Nat. Acad. Sci.*, 113, 32, 8927–8932 (2016).

Choi, Y. S., Cho, H., Ho, C. H., Lindzen, R. S., Park, S. K., & Yu, X. (2014). Influence of non-feedback variations of radiation on the determination of climate feedback. *Theoretical and applied climatology*, 115(1-2), 355-364.

He, J. and B. Soden, A re-examination of the projected subtropical precipitation decline, *Nature Climate Change*, doi:10.1038/nclimate3157 (2016).

Lau W. K.-M. & K.-M. Kim, Robust Hadley Circulation changes and increasing global dryness due to CO2 warming from CMIP5 model projections, *Proc. Nat. Acad. Sci.*, 112 (12), 3630–3635, doi: 10.1073/pnas.1418682112 (2015).

Mauritsen, T. & Stevens, B. Missing iris effect as a possible cause of muted hydrological change and high climate sensitivity in models. *Nature Geosci.* 8, 346–351 (2015).

Neelin, J. D., C. Chou, and H. Su, Tropical drought regions in global warming and El Nino teleconnections. *Geophys. Res. Lett.*, 30(24) 2275, doi:10.1029/2003GLO018625 (2003).

Neelin, J. D. and H. Su, Moist teleconnection mechanisms for the tropical South American and Atlantic sector. *J. Climate*, 18, 3928-3950, doi:10.1175/JCLI3517.1 (2005).

Su, H., J.H. Jiang, C. Zhai, T.J. Shen, J.D. Neelin, G.L. Stephens, and L.Y. Yung, Weakening and Strengthening Structures in the Hadley Circulation Change under Global Warming and Implications for Cloud Response and Climate Sensitivity, *J. Geophys. Res.*, 119, 10, 5787–5805, doi:10.1002/2014JD021642, (2014).

Reviewers' comments:

Reviewer #1 (Remarks to the Author):

Re-review of Su et al "Tightening of Hadley ascent key to radiative control on precipitation change in a warmer climate."

In my previous reviews I tried to distinguish between the circulation change associated with tropical SST variations associated with ENSO and the thermodynamic response of clouds with warming irrespective of structure. The authors have argued that it is the thermodynamic response of clouds to warming that is critical and not well represented by models and used AMIP experiments as evidence of this. They then argued that the structure of the SST response is not critical. The title and abstract still focus on the large-scale circulation response, mentioning the Hadley cell as a critical component. The most important conclusion of the paper, however, seems to be that the shrinkage of upper level tropical clouds with warming, and their effect on atmospheric cooling rates and thereby tropical precipitation is underestimated in most models and that therefore the sensitivity of tropical precipitation to surface temperature in nature is larger than most models predict. This is all somewhat conditioned on the assumption that observed ENSO variations are a good analog for global warming, a dubious thing. A more descriptive title then might be "Shrinkage of high tropical cloud area a key control on precipitation response to warming". But this is not a novel idea, so maybe it should be more specific still, "Most CMIP5 models underestimate shrinking of high tropical clouds and increase of precipitation in a warming climate."?

Some parts of the paper are poorly written and could be made much more succinct, I feel.

Specific Comments:

Lines 81-82: This is simply a constraint of the atmospheric energy balance, and so merely means that the models approximately conserve energy.

120-125: Here the authors indirectly invoke the SST distribution, which causes the mentioned circulation change, whereas in their response to review they argue that this is not important to their result.

130-132: It seems to me that the decreased longwave warming of clouds would enhance the atmospheric cooling rate and increase the upward motion in the convective regions. Heating of the atmosphere by cloud radiative effects actually decreases precipitation.

130-147: This piecemeal collection of partial physical arguments is confusing, and not entirely consistent.

177-180: Are you saying there are more clouds when the mean motion is upward? This is not new.

211-214: The tightening of the circulation in models is likely very sensitive to the convective parameterizations, and to the parameterizations that determine upper level ice cloud (anvil cloud). Global Climate models do not include many of the processes that maintain anvil clouds.

215-216: Arguably, the convection drives the upward motion and is not merely embedded in it.

284: I don't understand logically why the range of -2.39 to -1.43 is much larger than the uncertainty ranges given of ± 0.29 and ± 0.32 . Wouldn't it make more sense to just say -2.4 to -1.4?

287: It's a factor of 10 smaller, which is certainly significant.

295-296: Much of the observed signal is related to ENSO, which may largely be a change of SST shape as much as a change in mean temperature.

298-299: I don't believe that a linear regression on observed natural variability can be closely linked to climate feedbacks without many caveats, even if differences in natural variability can be linked to spread of feedbacks in models.

297 & 301: Too many significant figures are given in your slope estimates. 3.8 ± 0.4 and 4.0 ± 0.5 is adequate precision.

314-379: This section is dense with lots of regression coefficients differing by small amounts and at the end the overall conclusion is lost or muddled. This does not seem like breakthrough science.

388-390: Do we need to improve modeling of tropical circulation, or is it the simulation of upper level ice cloud that we need to improve. I think it is the latter, if I understood the main point of the paper.

Reviewer #3 (Remarks to the Author):

The authors have addressed my remaining questions with further analysis of temperature-mediated precipitation sensitivities although the quoted value of 2.1%/K to 3.2%/K (L361) is referred to as 2%/K to 3%/K on L42 and incorrectly as 1.1%/K to 0.3%/K on L393 so some careful checking of the manuscript is required. A clearer outline is provided of how robust physical processes across time-scales are identified from the regressions with surface temperature while the energy budget-based metrics provide the emergent constraints (notwithstanding the caveat that ENSO variability considered in a more simplistic way, is not a good surrogate for climate change). Aside from the missing reference list associated with supplementary Table 2 and a citation manager error on L319 and subject to the authors carefully checking the manuscript, I consider that the work is suitable for publication.

Response to Reviewers

Comments from the reviewers are marked in red and italicized. Our responses are in blue. The page numbers and line numbers in the response below refer to the clean version of the revised manuscript.

Summary:

We appreciate the reviewers' suggestions for improving the presentation of the paper. We have clarified the physical mechanisms according to Reviewer 1's comments and corrected the inconsistencies as Reviewer 2 pointed out. The manuscript has been reformatted to comply with the Nature Communications requirements. A point-by-point response is provided below.

Response to Reviewer 1

We thank the reviewer's detailed comments and useful suggestions. We have changed the title of the manuscript and rewritten the paragraphs about the interactions between circulation, clouds, radiation and precipitation (original Line 130-147). Additional analysis is performed to clarify the role of the Hadley Circulation change. All specific comments are addressed, and the corresponding changes are made in the manuscript.

"In my previous reviews I tried to distinguish between the circulation change associated with tropical SST variations associated with ENSO and the thermodynamic response of clouds with warming irrespective of structure. The authors have argued that it is the thermodynamic response of clouds to warming that is critical and not well represented by models and used AMIP experiments as evidence of this. They then argued that the structure of the SST response is not critical. The title and abstract still focus on the large-scale circulation response, mentioning the Hadley cell as a critical component. The most important conclusion of the paper, however, seems to be that the shrinkage of upper level tropical clouds with warming, and their effect on atmospheric cooling rates and thereby tropical precipitation is underestimated in most models and that therefore the sensitivity of tropical precipitation to surface temperature in nature is larger than most models predict. This is all somewhat conditioned on the assumption that observed ENSO variations are a good analog for global warming, a dubious thing. A more descriptive title then might be "Shrinkage of high tropical cloud area a key control on precipitation response to warming". But this is not a novel idea, so maybe it should be more specific

still, “Most CMIP5 models underestimate shrinking of high tropical clouds and increase of precipitation in a warming climate.”

We apologize that our previous response to the reviewer’s comments may have caused confusion. An itemized clarification is provided below. The relevant text is modified accordingly.

(1) We think the reduction of tropical ascending area and the decrease of high cloud fraction under global warming are governed by basic thermodynamics. As Reviewer 1 pointed out in the last review, a reduction of convective mass flux under global warming is expected from basic thermodynamics. Bony et al. (2016) showed the decrease of anvil cloud amount occurs under uniform SST warming in aqua-planet simulations under radiative-convective equilibrium. These theories do not require particular SST distributions. The interannual and centennial sensitivities are defined in a very simplistic way in this study: only the tropical-mean T_s change is considered, regardless of the shape of SST warming. For the interannual sensitivities, when we remove the strong El Niño year (June 97 to May 98) in the time series for regressions against T_s , all the relationships hold consistently with even higher correlations. Therefore, we presume that the key driver for the interannual and centennial similarity is the tropical-wide warming. However, we acknowledge that the exact role of SST spatial distributions needs further study. The El Niño-like SST warming pattern simulated in most climate models may contribute to the similar relations between interannual and centennial time scales. Based on Byrne and Schneider (2016), the meridional gradient of moisture and moist static energy are important for the narrowing of the ITCZ (defined by the time-mean zonal-mean mass stream function associated with upward vertical velocity at 700 hPa, which is equivalent to our definition of the width of ascending branch of the Hadley Cell). Moreover, the eastward shift of deep convection during El Niño may help to create a more zonally uniform ITCZ. We have added discussions in Line 176-185 and Line 358-361.

(2) The connection between the tightening of Hadley Circulation to the tropical-wide high cloud reduction is a new and important finding of this study. The basic thermodynamic arguments do not dictate any spatial structure of tropical circulation change when the overall convective mass flux is reduced under surface warming. For example, Figure 1 in Bony et al. (2016) (reproduced below) shows that randomly distributed convective systems would become aggregated with uniform surface warming, but there is no outstanding spatial structure (Earth’s rotation is set to zero in these simulations). However, in the model simulations with realistic land-ocean configurations and spatially varying mean atmospheric states, a tightening of Hadley ascent is

Fig. 1. Monthly precipitation (normalized by its global mean value) predicted by the IPSL, MPI, and NCAR GCMs in RCE simulations forced by an SST of (Top) 295 K and (Bottom) 305 K.

produced (e.g., Su et al. 2014). This suggests that the tightening of Hadley ascent or the narrowing of ITCZ is one dominant consequence that derives from the basic thermodynamics acting on the spatially varying climatological mean states. Such structural change of the Hadley Circulation can be explained by the thermodynamic theory involving the advection of moist static energy by mean circulation (e.g., Byrne and Schneider 2016; Neelin et al. 2003); however, the previous studies did not address the connection between such circulation change and tropical cloud fraction change. Our analyses (Figure 1a/b and Figure 2) show that the inter-model spread in the extent of the tightening of Hadley ascent accounts for about 40% of the across-model variance of the tropical-mean high cloud fraction decrease. To better relate the tropical-wide reduction of ascending area (the dF_{ω}/dT_s index used in the paper) to the Hadley Circulation, we have conducted additional analysis that shows the dF_{ω}/dT_s index is positively correlated with the change in the width of the ascending branch of the Hadley Circulation on both interannual and centennial time scales (new Supplementary Figure 7). This new analysis result demonstrates that the Hadley Circulation change is a critical component in the circulation response to surface warming. Corresponding discussions are added in Line 83-87 and Line 143-149.

(3) We agree with the reviewer that the most important quantity that drives the inter-model spread in precipitation sensitivity is the upper-level cloud fraction change. Our analysis on the circulation and cloud relation quantifies the role of circulation change in the model-spread of the high cloud fraction change. It suggests that constraining the model simulations of the circulation change would be an effective pathway towards improving the cloud simulations. For example, as the reviewer pointed out, circulation changes in the models are likely very sensitive to convective parameterizations. By constraining the circulation-sensitive parameters in the convective parameterizations, one would be able to narrow the model spread in cloud fraction simulations. The sentences in Line 161-167 and Line 349-354 describe these points.

(4) We agree with the reviewer that ENSO is not a good analogy of global warming in many aspects, evidenced by the different interannual and centennial sensitivities shown in our figures. However, the interannual and centennial variabilities share some common characteristics: the tropical-wide warming and the approximately similar SST warming pattern, although it is not clear how much the second feature contributes to the cloud and circulation sensitivities examined in the paper. The correlations between the inter-model spreads on the two time scales enable the use of the observed short-term variabilities to constrain the range of future predictions, as the reviewer concurred. We have added discussions about “ENSO is not a surrogate of global warming” in Line 185-188 and 358-361.

(5) Our study has three major findings:

- a) We demonstrate that the model differences in the tightening of Hadley ascent account for about 40% of the inter-model variance in the tropical-mean high cloud fraction decrease under global warming.

- b) We find that the relations between the circulation, cloud, OLR and precipitation responses to surface warming are strikingly similar between the interannual and centennial variations, and the relative magnitudes of precipitation sensitivity on the two time scales vary consistently across the models.
- c) We show that most CMIP5 models underestimate the interannual global-mean precipitation sensitivity because of muted “iris effect” using state-of-the-art satellite observations and infer that the hydrological sensitivity under global warming is likely on the higher end of CMIP5 model predictions.

These results have never been presented in a coherent manner in any previous studies; thus our study is an innovative contribution that would make a significant impact on advancing climate research. We have changed the title to be “Tightening of Hadley Ascent and Tropical High Cloud Region Key to Precipitation Change In a Warmer Climate” to emphasize both the circulation and cloud changes. We have also modified the text throughout the manuscript to make these points clearer.

Some parts of the paper are poorly written and could be made much more succinct, I feel.”

We have revised the paper substantially to be clearer and more succinct.

Specific comments:

Lines 81-82: This is simply a constraint of the atmospheric energy balance, and so merely means that the models approximately conserve energy.

Yes, the reviewer is right that the strong correlation between the inter-model spreads in dP/dT_s and $dLWC/dT_s$ shows that the models approximately conserve energy. This paragraph has been moved to the Supplementary Information and we have added a sentence “Corroborating that the climate models approximately conserve energy” in Line 23 in the Supplementary Information.

120-125: Here the authors indirectly invoke the SST distribution, which causes the mentioned circulation change, whereas in their response to review they argue that this is not important to their result.

As mentioned earlier, the SST distribution may be important in driving the narrowing of the Hadley ascent. The original sentence has been rewritten.

130-132: It seems to me that the decreased longwave warming of clouds would enhance the atmospheric cooling rate and increase the upward motion in the convective regions. Heating of the atmosphere by cloud radiative effects actually decreases precipitation.

You are right. Thanks for pointing it out. We have removed this sentence and made the paragraph more compact.

130-147: This piecemeal collection of partial physical arguments is confusing, and not entirely consistent.

This entire paragraph is shortened and rewritten. Please see the revised clean version Line 88 to 99.

177-180: Are you saying there are more clouds when the mean motion is upward? This is not new.

We have rephrased the sentence to be “it is known that high clouds are usually associated with upward motion” in Line 128.

211-214: The tightening of the circulation in models is likely very sensitive to the convective parameterizations, and to the parameterizations that determine upper level ice cloud (anvil cloud). Global Climate models do not include many of the processes that maintain anvil clouds.

Following the reviewer’s comment, we have revised the sentence and discussed convective parameterization and cloud parameterization. See the revised text in Line 161-174.

215-216: Arguably, the convection drives the upward motion and is not merely embedded in it.

We have rephrased the sentence. See the revised text in Line 197-198.

284: I don’t understand logically why the range of -2.39 to -1.43 is much larger than the uncertainty ranges given of ± 0.29 and ± 0.32 . Wouldn’t it make more sense to just say -2.4 to -1.4?

We have simplified the uncertainty range to “-2.4 to -1.4 %/K” in Line 246.

287: It’s a factor of 10 smaller, which is certainly significant.

Agree! We added “10 times smaller” in the sentence in Line 248.

295-296: Much of the observed signal is related to ENSO, which may largely be a change of SST shape as much as a change in mean temperature.

We agree with the reviewer and note both the change of SST shape and the change in mean temperature. See the revised text in Line 251-253.

298-299: I don’t believe that a linear regression on observed natural variability can be closely linked to climate feedbacks without many caveats, even if differences in natural variability can be linked to spread of feedbacks in models.

We have removed the mention of “climate feedbacks” in the sentence in Line 263.

297 & 301: Too many significant figures are given in your slope estimates. 3.8 ± 0.4 and 4.0 ± 0.5 is adequate precision.

Done as suggested.

314-379: This section is dense with lots of regression coefficients differing by small amounts and at the end the overall conclusion is lost or muddled. This does not seem like breakthrough science.

We have shortened the discussions in this section. Some detailed calculations are moved to the Supplementary Information. We believe the emergent constraint on the hydrological sensitivity based on the interannual variability is a novel result. No previous studies have shown the strong correlation between the inter-model spreads in interannual precipitation sensitivity to temperature-mediated hydrological sensitivity and no one has applied a robust observational estimate to constrain predictions of hydrological sensitivity from the longwave radiative balance.

388-390: Do we need to improve modeling of tropical circulation, or is it the simulation of upper level ice cloud that we need to improve. I think it is the latter, if I understood the main point of the paper.

We agree with the reviewer that improving the upper-level ice cloud is critically important for improving the hydrological sensitivity. Our study suggests that improving model physics that govern the tightening of Hadley ascent may be an effective pathway to improve high cloud simulations. In other words, model parameters pertaining to ice cloud physics and circulation change are probably equally important to cloud simulations and future climate predictions. See the revised text in Line 349-354.

Response to Reviewer 3

We appreciate the reviewer’s careful reading and suggestions. The reviewer’s comments have greatly helped us improve the quality of the paper.

The authors have addressed my remaining questions with further analysis of temperature-mediated precipitation sensitivities although the quoted value of 2.1%/K to 3.2%/K (L361) is referred to as 2%/K to 3%/K on L42 and incorrectly as 1.1%/K to 0.3%/K on L393 so some careful checking of the manuscript is required. A clearer outline is provided of how robust physical processes across time-scales are identified from the regressions with surface temperature while the energy budget-based metrics provide the emergent constraints (notwithstanding the caveat that ENSO variability considered in a more simplistic way, is not a good surrogate for climate change). Aside from the missing reference list associated with supplementary Table 2 and a citation

manager error on L319 and subject to the authors carefully checking the manuscript, I consider that the work is suitable for publication.

We have corrected the inconsistencies about the precipitation sensitivities. We have added the caveats that ENSO is not a good surrogate for climate change (see Line 185-188 and 358-361). The references listed in Table 2 are added in Supplementary Information, and the citation is corrected.

References cited above:

Bony, S, B. Stevens, D. Coppin, T. Becker, K. A. Reed, A. Voigt, and B. Medeiros, Thermodynamic control of anvil cloud amount. *Proc. Nat. Acad. Sci*, 113, 32, 8927–8932 (2016).

Byrne, M. P., and T. Schneider, Narrowing of the ITCZ in a warming climate: Physical mechanisms, *Geophys. Res. Lett.*, 43, 11,350–11,357, doi:10.1002/2016GL070396 (2016).

Neelin, J. D., C. Chou, and H. Su, Tropical drought regions in global warming and El Nino teleconnections. *Geophys. Res. Lett.*, 30(24) 2275, doi:10.1029/2003GLO018625 (2003).

Su, H., J.H. Jiang, C. Zhai, T.J. Shen, J.D. Neelin, G.L. Stephens, and L.Y. Yung, Weakening and Strengthening Structures in the Hadley Circulation Change under Global Warming and Implications for Cloud Response and Climate Sensitivity, *J. Geophys. Res.*, 119, 10, 5787–5805, doi:10.1002/2014JD021642, (2014).

REVIEWERS' COMMENTS:

Reviewer #3 (Remarks to the Author):

The authors have addressed my suggestions and I consider that the responses adequately address most of the remaining comments by reviewer 1. I just have some final minor suggestions.

1) Title: This is fine but a shorter version, based on Reviewer 1 comments, could be "Tightening of tropical high cloud regime key to precipitation change in a warmer climate" (or slightly longer "Tightening of tropical ascending regime and high cloud key to precipitation change in a warmer climate")

2) Abstract - I suggest the authors check carefully for clarity. A suggestion for the 2nd line is: "Here we show that tightening of the ascending branch of the Hadley Circulation coupled with decreases in tropical high cloud fraction is key in modulating precipitation response to surface warming."

2) The response "it is not clear how much the second feature contributes to the cloud and circulation sensitivities examined in the paper" is clearly important, particularly if future regional responses are not robust and this could be an additional caveat e.g. L253 (the discussion L175-188 is good I think)

3) The response to Reviewer 1: "the strong correlation between the inter-model spreads in dP/dTs and $dLWC/dTs$ shows that the models approximately conserve energy." is not correct as LWC does not include sensible heat and shortwave absorption. Nevertheless I agree that it is not a new finding.

4) L164 check use of commas e.g. "for example[,] the entrainment rate"

5) L280 - it would be useful to remind the reader that high clouds are central to the "iris effect"

6) L223 "On the one hand, the decrease of high cloud amount reduces the cloud longwave warming effect to the Earth-atmosphere system. On the other hand, the shrinkage of high cloud cover enlarges the dry and clear areas through which lower tropospheric thermal emissions escape to space. Both effects enhance negative longwave radiative feedback."

This is the same thing isn't it? I suggest shortening:

"The decrease of high cloud amount reduces the cloud longwave warming effect on the Earth-atmosphere system, enlarging the dry and clear areas through which lower tropospheric thermal emissions escape to space and enhancing the negative longwave radiative feedback."

L356 "consistently across" - the emphasis seems in the wrong place here and I think you mean to emphasise this relationship across models as opposed to between the time-scales for individual models.

Response to Reviewers

Comments from the reviewer are marked in red and italicized. Our responses are in blue. The line numbers in the responses below refer to the clean version of the revised manuscript.

Summary:

We greatly appreciate the detailed suggestions by the reviewer. We have modified the manuscript accordingly with the point-by-point response below.

Response to Reviewers

The authors have addressed my suggestions and I consider that the responses adequately address most of the remaining comments by reviewer 1. I just have some final minor suggestions.

1) Title: This is fine but a shorter version, based on Reviewer 1 comments, could be "Tightening of tropical high cloud regime key to precipitation change in a warmer climate" (or slightly longer "Tightening of tropical ascending regime and high cloud key to precipitation change in a warmer climate")

Following the reviewer's suggestion, we change the title to "Tightening of tropical ascent and high clouds key to precipitation change in a warmer climate".

2) Abstract - I suggest the authors check carefully for clarity. A suggestion for the 2nd line is:

"Here we show that tightening of the ascending branch of the Hadley Circulation coupled with decreases in tropical high cloud fraction is key in modulating precipitation response to surface warming."

Done as suggested.

2) The response "it is not clear how much the second feature contributes to the cloud and circulation sensitivities examined in the paper" is clearly important, particularly if future regional responses are not robust and this could be an additional caveat e.g. L253 (the discussion L175-188 is good I think)

The caveat associated with the SST warming pattern is discussed in L158-162, L177-186, L251-253 and L359-362.

3) The response to Reviewer 1: "the strong correlation between the inter-model spreads in dP/dTs and $dLWC/dTs$ shows that the models approximately conserve energy." is not correct as LWC does not include sensible heat and shortwave absorption. Nevertheless I agree that it is not a new finding.

We agree with the reviewer and have removed the sentence in the second line of Supplementary Information.

4) L164 check use of commas e.g. "for example[,] the entrainment rate"

Comma removed.

5) L280 - it would be useful to remind the reader that high clouds are central to the "iris effect"

We have added "of high clouds" in the sentence.

6) L223 "On the one hand, the decrease of high cloud amount reduces the cloud longwave warming effect to the Earth-atmosphere system. On the other hand, the shrinkage of high cloud cover enlarges the dry and clear areas through which lower tropospheric thermal emissions escape to space. Both effects enhance negative longwave radiative feedback."

This is the same thing isn't it? I suggest shortening:

"The decrease of high cloud amount reduces the cloud longwave warming effect on the Earth-atmosphere system, enlarging the dry and clear areas through which lower tropospheric thermal emissions escape to space and enhancing the negative longwave radiative feedback."

Done as suggested. Thank you!

7) L356 "consistently across" - the emphasis seems in the wrong place here and I think you mean to emphasize this relationship across models as opposed to between the time-scales for individual models.

The sentence is modified as "the relative magnitudes of the OLR and precipitation sensitivities to surface warming vary consistently across the models on the interannual and centennial time scales".